# Novel candidates of pathogenic variants of the *BRCA1* and *BRCA2* genes from a dataset of 3,552 Japanese whole genomes (3.5KJPNv2)

**Hideki Tokunaga**[1,2], **Keita Iida**[3,4], **Atsushi Hozawa**[3], **Soichi Ogishima**[2,3], **Yoh Watanabe**[5], **Shogo Shigeta**[1], **Muneaki Shimada**[1,2], **Yumi Yamaguchi-Kabata**[3], **Shu Tadaka**[3], **Fumiki Katsuoka**[3,6], **Shin Ito**[3,7,8], **Kazuki Kumada**[2,3], **Yohei Hamanaka**[3], **Nobuo Fuse**[3], **Kengo Kinoshita**[2,3,7,8,9], **Masayuki Yamamoto**[2,3,6], **Nobuo Yaegashi**[1,3], **Jun Yasuda**[3,7,8]*

1 Department of Obstetrics and Gynecology, Tohoku University Graduate School of Medicine, Sendai, Japan, 2 Advanced Research Center for Innovations in Next-Generation Medicine, Tohoku University, Sendai, Japan, 3 Tohoku Medical Megabank Organization, Tohoku University, Sendai, Japan, 4 Laboratory of Cell Systems, Institute for Protein Research, Osaka University, Suita, Osaka, Japan, 5 Department of Obstetrics and Gynecology, Tohoku Medical and Pharmaceutical University, Sendai, Japan, 6 Department of Medical Biochemistry, Tohoku University Graduate School of Medicine, Sendai, Japan, 7 Department of Applied Information Sciences, Graduate School of Information Sciences, Tohoku University, Sendai, Japan, 8 Division of Molecular and Cellular Oncology, Miyagi Cancer Center Research Institute, Natori, Miyagi, Japan, 9 Institute of Development, Aging, and Cancer, Tohoku University, Sendai, Japan

* jun-yasuda@miyagi-pho.jp

**Data Availability Statement:** In terms of the ethical restrictions on access to the data used in our study, the data that we used are histories of

## Abstract

Identification of the population frequencies of definitely pathogenic germline variants in two major hereditary breast and ovarian cancer syndrome (HBOC) genes, *BRCA1/2*, is essential to estimate the number of HBOC patients. In addition, the identification of moderately penetrant HBOC gene variants that contribute to increasing the risk of breast and ovarian cancers in a population is critical to establish personalized health care. A prospective cohort subjected to genome analysis can provide both sets of information. Computational scoring and prospective cohort studies may help to identify such likely pathogenic variants in the general population. We annotated the variants in the *BRCA1* and *BRCA2* genes from a dataset of 3,552 whole-genome sequences obtained from members of a prospective cohorts with genome data in the Tohoku Medical Megabank Project (TMM) with InterVar software. Computational impact scores (CADD_phred and Eigen_raw) and minor allele frequencies (MAFs) of pathogenic (P) and likely pathogenic (LP) variants in ClinVar were used for filtration criteria. Familial predispositions to cancers among the 35,000 TMM genome cohort participants were analyzed to verify the identified pathogenicity. Seven potentially pathogenic variants were newly identified. The sisters of carriers of these moderately deleterious variants and definite P and LP variants among members of the TMM prospective cohort showed a statistically significant preponderance for cancer onset, from the self-reported cancer history. Filtering by computational scoring and MAF is useful to identify potentially pathogenic variants in *BRCA* genes in the Japanese population. These results should help to follow up the carriers of variants of uncertain significance in the HBOC genes in the longitudinal prospective cohort study.

disease and genomic information; both of these sets of data are private and it would be possible to identify an individual with them. Therefore, it is necessary to obtain approval for data access from the TMM prospective cohort project; specifically, users should obtain approval from the sample and data access committee of the TMM Biobank. This committee consists of experts both inside and outside the TMM. Upon applying to this committee, the Group of Materials and Information Management in the TMM at Tohoku University supports the procedures for data transfer. The Group of Materials and Information Management can be contacted at dist@megabank.tohoku.ac.jp.

**Funding:** This work was supported by JSPS KAKENHI (Grant Number JP17K07193, JP19H03795, and JP17K11265) for JY, NY, and MS, respectively. This work was supported by The National Cancer Center Research and Development Fund (29-A-3) and AMED (Grant Number JP19ck0106319) for NY and HT, respectively. This work was supported in part by the Tohoku Medical Megabank Project through the Ministry of Education, Culture, Sports, Science and Technology (MEXT), Japan for MY; the Reconstruction Agency, MEXT, Japan for MY; by the Japan Agency for Medical Research and development (AMED; Grant numbers JP17km0105001 and JP17km0105002) awarded to MY; and AMED GRIFIN project (grant numbers JP17km0405203 and JP18km0405203) awarded to MY. All computational resources were provided by the ToMMo supercomputer system (http://sc.megabank.tohoku.ac.jp/en), which is supported by the Facilitation of R&D Platform for AMED Genome Medicine Support conducted by AMED (Grant number JP17km0405001) awarded to MY.

**Competing interests:** The authors have declared that no competing interests exist.

## Introduction

Since the precision medicine initiative was launched in 2015 by the US government [1], prediction of the disease risks of individuals by using their genomic information has become plausible in a clinical setting. In Japan, gene profiling assays for cancer tissues and companion diagnostic tests for cancer-predisposing genes are now covered by the national health insurance system. These gene profiling tests can examine variations in most of the genes conferring susceptibility to two major adult-onset hereditary cancer-predisposing syndromes, hereditary breast and ovarian cancer syndrome (HBOC) and Lynch syndrome. Nowadays, the clinical significance of variants of these genes is important for patient care and the health of their relatives at the bedside.

The correct judgment of the pathogenicity of germline variants in these cancer-predisposing genes is critical for the physicians who manage such patients and undertake gene profiling analyses for cancer treatment. For example, in carriers of disease-causing mutations of HBOC, prophylactic surgery is beneficial [2, 3]. Testing of *BRCA* genes may help carriers' decision-making regarding prophylactic salpingectomy or salpingo-oophorectomy because, in patients with high-grade serous carcinoma arising from the fallopian tube, germline *BRCA* mutations are more prevalent in Japanese women than in other ethnic groups [4, 5]. Synthetic lethal drugs for cancers associated with homologous recombination defects are available for patients carrying disease-causing mutations of HBOC [6, 7]. In this context, variants of uncertain significance (VUSs) would clearly be a source of major problems for clinicians. Kurian et al. reported that inexperienced breast surgeons tend to manage patients with VUSs in the *BRCA1* or *BRCA2* gene as pathogenic HBOC mutation carriers [8]. This means that the lack of comprehensive annotation methods for variants might cause overdiagnosis or overtreatment in patients with *BRCA* mutations that are uncharacterized but actually benign.

To overcome these difficulties, several levels of studies (single organization, single nation, and whole world level) have been done previously. As a single organization study, Sugano et al. reported the *BRCA1* and *BRCA2* germline variants in 135 HBOC patients and identified 28 pathogenic ones [9]. As the nationwide study, Arai et al. examined 830 Japanese HBOC pedigrees collected by the Japanese HBOC consortium and identified 49 different pathogenic variants among them [10]. Similarly, a nationwide multicenter study revealed that germline *BRCA 1/2* mutations were present in 14.7% of 634 Japanese women with ovarian cancer [5]. Lee et al. also examined the variants in the *BRCA1* and *BRCA2* genes in breast and ovarian cancer patients' germline genomic DNA and calculated posterior probabilities for the disease-causing mutations; they identified five previously unreported variants as candidate pathogenic ones [11]. Finally, as an international study, the BRCA Challenge project established an open access database, BRCA Exchange for providing reliable and easily accessible variant data for better clinical treatments of HBOC [12]. As of October 2020, the BRCA Exchange database has collected more than 40,000 variants in the BRCA1/2 genes from major clinical databases and estimated their pathogenicity under expert peer review in collaboration with the ENIGMA consortium [13]. The purposes of this comprehensive database are to provide reliable and easily accessible variant data interpreted for the high-penetrance phenotype of HBOC and to develop a model database for the utilization and sharing of public data to provide better clinical treatments of hereditary disease. In this database, there are more than 4,900 variants annotated as "pathogenic" by the ENIGMA consortium. Recently, a large-scale Japanese project involving the sequencing of HBOC patients' germline genomic DNA for 11 breast cancer-predisposing genes revealed 134 pathogenic germline variants concentrated in cancer patients in the *BRCA1* and *BRCA2* genes [14]. Patient-based studies for identifying germline pathogenic variants are very effective for identifying potential variants of this kind, but cannot estimate

the frequencies of those alleles in the general population, which is critical for estimating the number of HBOC patients in a community. In addition, moderately deleterious HBOC gene variants contribute to increase the risk of breast and ovarian cancers in a population, so identifying them is critical for establishing personalized health care. The carriers of moderately deleterious HBOC variants would not undergo drastic prophylactic modalities, but frequent examination would be recommendable for earlier detection of the cancers. A prospective cohort subjected to genome analysis would provide both sets of information.

Only analyses of prospective cohorts of the general population can confirm the causality of VUSs via the collection of follow-up data and using the precise minor allele frequencies. However, in the case of follow-up surveys in prospective cohorts, it is critical to focus on the participants who need to be carefully followed up because of the limitation of available resources [15]. An appropriate method to select participants for detailed follow-up studies is critical for analyzing the causalities of germline VUSs in cancer-predisposing genes.

Here, we describe the levels of known and potentially disease-causing variants in the *BRCA* genes among the general Japanese population, by analyzing a whole-genome reference panel for the Japanese characterized by the Tohoku Medical Megabank (TMM) Project. The TMM Project involves a combination of prospective cohort, biobanking, and genome-omics analysis (for reviews, see [16–19]). The dataset collected so far includes more than 3,500 independent whole-human-genome sequences (3.5KJPNv2) [20] with self-reported individual and family history data. The main benefit of the whole-genome sequencing of the dataset is that it provides more comprehensive information of the structure of the two HBOC genes than the exome-based approach. We also refer to these data to test whether computational annotation can identify any variants that might cause HBOC with high penetrance.

## Materials and methods

### Ethics approval and consent to participate

This study was approved by the ethics committee of Tohoku Medical Megabank Organization at Tohoku University (registration number: 2018-4-003). All participants in the present study were recruited by Tohoku Medical Megabank Organization at Tohoku University and provided written informed consent to participate in the cohort study.

### Dataset

Subjects were obtained from the TMM Community-Based Cohort (TMM CommCohort) Study established by Tohoku Medical Megabank Project [21], in which more than 120,000 adults participate. The whole-genome sequences of some of the participants have been obtained; the criteria for selecting WGS samples are described elsewhere [19, 22]. In brief, the samples for development of the Japanese whole-genome sequencing dataset were selected based on the SNP array data of the samples. Only one sample was picked up from a kinship group to obtain the precise allele frequencies. The whole-genome sequencing was performed with HiSeq 2500 sequencers (Illumina, Inc., San Diego, CA) with a PCR-free protocol from the genomic DNA extracted from whole blood.

### Annotation of genomic variants in the *BRCA* genes

The 3.5KJPNv2 variant data were downloaded from the jMorp database (https://jmorp.mega bank.tohoku.ac.jp/) [23]. The dataset is divided in two in terms of autosomal variants, namely, single-nucleotide variations (tommo-3.5KJPNv2v2-20181105open-af_snvall-autosome.vcf.gz) and indels (tommo-3.5KJPNv2v2-20181105open-af_indelall-autosome.vcf.gz), with index

files. We defined the *BRCA1* and *BRCA2* regions based on GeneCards (https://www.gene cards.org/) [24] as chr17:41,196,312–41,277,500 and chr13:32,889,611–32,973,809 (hg19), respectively. Variant extraction was performed with bcftools [25, 26]. The 3.5KJPNv2 VCF file integrates multiple alleles in single lines, so normalization was performed with bcftools.

The *BRCA* variants in 3.5KJPNv2 were annotated with the InterVar [27] command line package (default options), which depends on ANNOVAR [28]. InterVar is an analytical package to estimate the clinical impact of gene variants based on guidelines for variant interpretation, namely, the American College of Medical Genomic Guidelines and those of the Association for Molecular Pathology in clinical sequencing [29]. InterVar includes annotations of ClinVar (version from December 1, 2015) [30] and predicts pathogenicity, using indices such as Combined Annotation Dependent Depletion (CADD) [31], DANN [32], and Eigen [33]. The positions of the candidate pathological variants found in the Korean population [11] were described as the cDNA positions. To apply the data to the InterVar software, the TransVar annotation program [34] was used to obtain the genomic positions of the variants, followed by the InterVar annotation described above. To compare the variant frequencies in 3.5KJPNv2 and in the gnomAD database for the *BRCA1/2* variants, we downloaded gnomAD data [35] from the associated webpage (https://gnomad.broadinstitute.org/; downloaded on February 23, 2020). The selected variants were visualized with the mutation mapper at cBioPortal [36, 37].

The RIKEN 2000 genome allele frequency data [38] were downloaded from the Japanese Encyclopedia of Genetic Associations (http://jenger.riken.jp/data) and TCGA germline variant data were as described previously [39].

## Obtaining individual and family histories

TMM prospective cohort project data are stored in a supercomputer system, with secure data access [40]. The TMM database is a relational database and it consists of several separate datasets. The key is the participants' IDs to link the information stored in the different tables. The individual and family histories were extracted from a large data matrix consisting of self-reported findings from a paper-based questionnaire given to the members of the cohort. The dataset consists of 35,199 participants in the TMM CommCohort and the data were frozen for distribution to the Japanese scientific community in 2017, as a provisional version. For most of the participants, whole-genome sequencing data are not available. The detailed method for obtaining the participants' past and family histories, which consist of 269 entries for malignant neoplasms and 1271 for other diseases, is described elsewhere [21]. We did not use TMM Project Birth and Three-Generation Cohort data because the participants are expected to be relatively young and their family members may not be old enough to obtain positive cases [41].

The self-reported questionnaire data were filtered out for the participants who checked more than 50 items for past and family histories of malignant neoplasms. Most of the participants who checked more than 50 items showed contradictory histories, such as a self-history of ovarian cancer being recorded by male participants. Therefore, we decided to remove such records and obtained 35,136 records as a result. In the statistical analysis comparing carriers of candidate *BRCA* pathogenic variants and other TMM CommCohort participants regarding self-reported individual and family histories, we employed the binomial distribution to calculate the p-value. Then, we calculated the accumulation of past and family histories only for the items of malignant neoplasms. The questionnaire just asked about the presence or absence of such histories, which could be represented as "0" or "1" for each item. This made it impossible to give a weight to the numbers of affected siblings or offspring.

In terms of the access to data from the TMM prospective cohort project, users should obtain approval from the sample and data access committee of the TMM Biobank [17]. This

committee consists of experts both inside and outside the TMM. Upon the receipt of an application to the committee, the Group of Materials and Information Management in the TMM at Tohoku University supports the procedures for data utilization.

### Statistics

To analyze the correlations among the three computational estimates of the impacts of variants, we employed R 3.6.1 for calculating the Pearson correlation coefficient. We applied Fisher's exact test and chi-squared test with Yates' correction for calculating the p-values of the differences in numbers of cancer-bearing family members.

## Results and discussion

### Summary of *BRCA* variants in 3.5KJPNv2

More than 3,600 variants were found in the *BRCA* genes, 6.15% of which are in coding regions. The total proportion of coding exonic regions of the two genes is 9.58% in hg19 and 23.1% of the total variants in the two genes in 3.5KJPNv2 are indels. Indel calling using the short-read sequence data is less reliable than the findings for single-nucleotide variants, so the indels found in 3.5KJPNv2 may require further verification using long-read sequencing data.

How many known pathogenic mutations of the *BRCA* genes are identified in 3.5KJPNv2? We estimated this in a previous study on 2KJPN [42], relative to which there should be more pathogenic variants here. S1 Table indicates the annotation results of the variants in the *BRCA1* and *BRCA2* regions using the InterVar package. Ten variants in the *BRCA* genes are annotated as "pathogenic" (P) or "likely pathogenic" (LP) by referring to the ClinVar database. The accumulated frequency of pathogenic variations of *BRCA* genes in 3.5KJPNv2 is 0.0018, which might be lower than the clinical estimation of HBOC carriers in Japan.

To obtain deeper insight into the 3.5KJPNv2 *BRCA* variants, we compared the results with the gnomAD database, which contains more than 130,000, multi-ethnic, human exome variants (https://gnomad.broadinstitute.org/) (S1 Table). The total number of pathogenic variants in the *BRCA* genes is much smaller in 3.5KJPNv2 than in gnomAD (S1 Table). However, considering that the numbers of collected samples are quite different, the numbers of P and LP variants in 3.5KJPNv2 per population were very similar to those for gnomAD. Specifically, the rates of ClinVar P or LP variants were $2.81 \times 10^{-3}$/person and $2.50 \times 10^{-3}$/person in 3.5KJPNv2 and gnomAD, respectively. To investigate the population specificity, we extracted ClinVar and InterVar P or LP variants found in East Asian populations in the gnomAD database (gnomAD-EAS; Table 1). Intriguingly, there were only four overlaps between 3.5KJPN and gnomAD-EAS for ClinVar and InterVar P or LP variants (Table 1). For example, one of the most prominent *BRCA1* pathogenic variants, L63X [9, 10], does not appear in gnomAD-EAS (Table 1). In contrast, the two most prevalent P or LP variants, BRCA2 p.G2508S and p.A2786T, are present in 3.5KJPNv2. These two variants may be commonly distributed among East Asian populations. These results support the notion that pathogenic variants of a gene are highly specific to each ethnic group and thus that population-specific collection of whole-genome sequencing data is critical for nationwide public health care planning [19].

### Estimation of pathogenic variants in the two *BRCA* genes in the Japanese population

In the case of ClinVar, the data are based on previous reports of the identification of pathogenic variants in disease-predisposed families, so there might be new, unreported pathogenic

**Table 1. P or LP variants in the *BRCA1/2* genes in gnomAD-EAS population.**

| Chr | Start | End | Ref | Alt | Gene. refGene | Func.refGene | AAChange | Clinvar | InterVar and Evidence | 3.5KJPN | gnomAD EAS |
|-----|-------|-----|-----|-----|---------------|--------------|----------|---------|----------------------|---------|------------|
| 13 | 32890557 | 32890558 | AG | - | *BRCA2* | splicing | NA | Likely_pathogenic | Uncertain significance | . | 1.09.E-04 |
| 13 | 32890627 | 32890627 | - | T | *BRCA2* | frameshift insertion | p.T10fs | Pathogenic | Pathogenic | . | 6.41.E-04 |
| 13 | 32905124 | 32905127 | GACA | - | *BRCA2* | frameshift deletion | p.V250fs | Pathogenic | Pathogenic | . | 5.44.E-05 |
| 13 | 32906565 | 32906565 | - | A | *BRCA2* | frameshift insertion | p.T317fs | Pathogenic | Pathogenic | . | 5.87.E-05 |
| 13 | 32907014 | 32907014 | A | T | *BRCA2* | stopgain | p.K467X | Pathogenic | Pathogenic | . | 5.53.E-05 |
| 13 | 32910831 | 32910831 | C | G | *BRCA2* | stopgain | p.S780X | Pathogenic | Pathogenic | . | 5.44.E-05 |
| 13 | 32910932 | 32910932 | C | - | *BRCA2* | frameshift deletion | p.P814fs | Pathogenic | Pathogenic | . | 5.51.E-05 |
| 13 | 32911601 | 32911601 | C | T | *BRCA2* | stopgain | p.Q1037X | Pathogenic | Pathogenic | . | 5.45.E-05 |
| 13 | 32911659 | 32911662 | AAAA | - | *BRCA2* | frameshift deletion | p.Q1056fs | Pathogenic | Pathogenic | . | 5.48.E-05 |
| 13 | 32912090 | 32912091 | TG | - | *BRCA2* | frameshift deletion | p.C1200fs | Pathogenic | Pathogenic | . | 5.44.E-05 |
| 13 | 32912234 | 32912237 | AGTG | - | *BRCA2* | frameshift deletion | p.S1248fs | Pathogenic | Pathogenic | . | 5.47.E-05 |
| 13 | 32913656 | 32913657 | AG | - | *BRCA2* | frameshift deletion | p.S1722fs | Pathogenic | Pathogenic | . | 6.42.E-04 |
| 13 | 32914066 | 32914069 | AATT | - | *BRCA2* | frameshift deletion | p.T1858fs | Pathogenic | Pathogenic | 0.0003 | 5.51.E-05 |
| 13 | 32914137 | 32914137 | C | A | *BRCA2* | stopgain | p.S1882X | Pathogenic | Pathogenic | . | 5.45.E-05 |
| 13 | 32914172 | 32914172 | - | A | *BRCA2* | stopgain | p. Y1894_E1895delinsX | Pathogenic | Pathogenic | . | 5.44.E-05 |
| 13 | 32914356 | 32914356 | C | A | *BRCA2* | stopgain | p.S1955X | Pathogenic | Pathogenic | . | 1.00.E-04 |
| 13 | 32914976 | 32914977 | AA | - | *BRCA2* | frameshift deletion | p.K2162fs | Pathogenic | Pathogenic | . | 5.77.E-05 |
| 13 | 32929367 | 32929370 | AAAC | - | *BRCA2* | frameshift deletion | p.K2459fs | Pathogenic | Pathogenic | . | 5.44.E-05 |
| 13 | 32930609 | 32930609 | C | T | *BRCA2* | stopgain | p.R2494X | Pathogenic | Pathogenic | . | 1.63.E-04 |
| 13 | 32931957 | 32931957 | - | A | *BRCA2* | frameshift insertion | p.D2566fs | Pathogenic | Pathogenic | . | 5.44.E-05 |
| 13 | 32937315 | 32937315 | G | T | *BRCA2* | splicing | NA | Pathogenic/ Likely_pathogenic | Pathogenic | . | 6.41.E-04 |
| 13 | 32937362 | 32937362 | A | G | *BRCA2* | nonsynonymous SNV | p.I2675V | Pathogenic/ Likely_pathogenic | Likely pathogenic | 0.0001 | 5.44.E-05 |
| 13 | 32913569 | 32913569 | - | T | *BRCA2* | frameshift insertion | p.L1693fs | UNK | Likely pathogenic | . | 5.71.E-05 |
| 13 | 32930651 | 32930651 | G | A | *BRCA2* | nonsynonymous SNV | p.G2508S | Conflicting | Likely pathogenic | 0.0003 | 2.26.E-03 |
| 13 | 32930727 | 32930727 | C | T | *BRCA2* | nonsynonymous SNV | p.S2533F | UNK | Likely pathogenic | . | 5.44.E-05 |
| 13 | 32932057 | 32932057 | A | C | *BRCA2* | nonsynonymous SNV | p.E2599A | UNK | Likely pathogenic | . | 5.44.E-05 |
| 13 | 32936755 | 32936755 | T | A | *BRCA2* | nonsynonymous SNV | p.M2634K | UNK | Likely pathogenic | . | 5.44.E-05 |
| 13 | 32937581 | 32937581 | G | C | *BRCA2* | nonsynonymous SNV | p.G2748R | UNK | Likely pathogenic | . | 5.44.E-05 |
| 13 | 32944557 | 32944557 | C | T | *BRCA2* | nonsynonymous SNV | p.R2784W | Conflicting | Likely pathogenic | . | 5.44.E-05 |
| 13 | 32944563 | 32944563 | G | A | *BRCA2* | nonsynonymous SNV | p.A2786T | Conflicting | Likely pathogenic | 0.0001 | 7.52.E-04 |
| 13 | 32953474 | 32953474 | G | T | *BRCA2* | nonsynonymous SNV | p.Q2925H | UNK | Likely pathogenic | . | 5.45.E-05 |
| 13 | 32953617 | 32953617 | G | A | *BRCA2* | nonsynonymous SNV | p.R2973H | Conflicting | Likely pathogenic | . | 1.64.E-04 |
| 13 | 32968844 | 32968844 | A | G | *BRCA2* | nonsynonymous SNV | p.Y3092C | Conflicting | Likely pathogenic | . | 5.02.E-05 |

(*Continued*)

**Table 1.** (Continued)

| Chr | Start | End | Ref | Alt | Gene. refGene | Func.refGene | AAChange | Clinvar | InterVar and Evidence | 3.5KJPN | gnomAD EAS |
|---|---|---|---|---|---|---|---|---|---|---|---|
| 13 | 32969032 | 32969032 | T | - | *BRCA2* | frameshift deletion | p.F3155fs | UNK | Likely pathogenic | . | 6.42.E-04 |
| 17 | 41201209 | 41201209 | G | - | *BRCA1* | frameshift deletion | p.Q1779fs | Pathogenic | Pathogenic | . | 5.44.E-05 |
| 17 | 41209095 | 41209095 | G | A | *BRCA1* | stopgain | p.R1751X | Pathogenic | Uncertain significance | . | 5.01.E-05 |
| 17 | 41215948 | 41215948 | G | A | *BRCA1* | nonsynonymous SNV | p.R1699W | Pathogenic | Uncertain significance | . | 5.44.E-05 |
| 17 | 41244106 | 41244106 | C | - | *BRCA1* | frameshift deletion | p.E1148fs | Pathogenic | Pathogenic | . | 5.44.E-05 |
| 17 | 41245115 | 41245115 | G | - | *BRCA1* | frameshift deletion | p.P811fs | Pathogenic | Pathogenic | . | 5.44.E-05 |
| 17 | 41256190 | 41256190 | G | T | *BRCA1* | stopgain | p.Y130X | Pathogenic | Pathogenic | . | 5.44.E-05 |
| 17 | 41276080 | 41276080 | G | A | *BRCA1* | stopgain | p.Q12X | Pathogenic | Pathogenic | . | 5.44.E-05 |

variants to be found in the general population. To address this issue, we applied an annotation approach with InterVar. As stated above, InterVar is designed to estimate the clinical importance of human genetic variants that have not been reported previously, in accordance with the American College of Medical Genetics (ACMG) Guidelines of secondary findings in clinical sequencing [29]. Interestingly, the package annotates another 13 variants as P or LP in the *BRCA* genes, as well as all of the 10 ClinVar P and/or LP variants. Among the 13 newly annotated P or LP variants, 4 are frameshift indels and 9 are nonsynonymous variants. None of these four frameshift indels is annotated with dbSNP, so it should not be considered as discordant with ClinVar. Four nonsynonymous variants detected by InterVar are annotated as "conflicting interpretation of pathogenicity" in the ClinVar database. One of the LP variants from InterVar, *BRCA1* p.L52F, shows quite high minor allele frequency (MAF) in 3.5KJPNv2 (0.0037) compared with other definite ClinVar P or LP variants. This variant was estimated to be a VUS in the Japanese HBOC consortium study [10] and "likely benign" by Lee et al. in a Korean prospective study on breast cancer patients.

There is a large publicly available dataset of Japanese whole-genome sequencing data from RIKEN [38]. It consists of deep sequencing data from 2,234 whole genomes (average depth of 25×), 1,939 of which are from BioBank Japan (BBJ), a large biobank of patients suffering from more than 50 diseases [43]. The detailed composition of the samples from BBJ is not available, but 1,276 patients with six diseases including breast cancer are included. Hence, it can be expected that pathogenic variants found in 3.5KJPNv2 might be enriched in the RIKEN dataset, although the selection criteria of the samples for the RIKEN project are unknown. As expected, two InterVar P or LP variants, *BRCA2* c.5573_5577C and *BRCA1* p.L63X, are enriched (9.75- and 16.2-fold, respectively) in the RIKEN dataset (S2 Table). In addition, a pathogenic variant not found in TMM 3.5KJPNv2 was identified (*BRCA2* p.E2877X). In contrast, the prevalent InterVar LP variant, *BRCA1* p.L52F, is not enriched in the RIKEN Japanese whole-genome dataset (0.5-fold, S2 Table). Similarly, we checked the germline variants of the *BRCA* genes in TCGA dataset [39] and found three *BRCA2* pathogenic variants that overlapped with 3.5KJPNv2 (p.T219fs, p.T1858fs, and p.N2134fs); all three of these are highly enriched in TCGA (382-fold, 318-fold, and 95.5-fold, respectively).

These results suggest that the annotation by InterVar may include false positives as well as false negatives, although no VUSs identified by the Japanese HBOC consortium are included in our estimation [10]. Precise data on the MAFs obtained by the unbiased selection of panel constituents from the general population are critical for estimating the pathogenicity of VUSs

and should be included in the criteria of pathogenicity for adult-onset hereditary disorders such as HBOC based on the InterVar annotation.

## Estimate of computational scoring tools' performance in predicting pathogenicity of novel 3.5KJPNv2 BRCA variants

InterVar annotates the variants' functional impact based on the ACMG guidelines and it largely depends on previous reports to define the parameters for scoring. For example, criterion PS1 of InterVar states that "the variant involves the same amino acid change as a previously established pathogenic variant regardless of nucleotide change." This means that one needs previous knowledge about pathogenic variants in order to annotate a variant as "pathogenic" by InterVar. In contrast, only one supportive item, PP3, is used from computational estimations in InterVar: "Multiple lines of computational evidence support a pathogenic effect on the gene or gene product (e.g., conservation, evolutionary, splicing impact). Hence, InterVar may underestimate the clinical impact of potentially pathogenic variants about which previous information is not available. The tendency might be worse in the noncoding regions in the coding genes like the *BRCA1/2* genes because of the lack of functional studies for such regions. Nowadays, the whole genome sequencing data is accumulating and comparisons between the phenotypes and variants in the noncoding regions found by the WGS will provide critical data for the interpretation of the noncoding variants. Therefore, we would like to test whether the unbiased, computational estimations of the pathogenicity of the variants can be used to find potentially pathogenic variants without previous knowledge.

The Pearson correlation coefficients of CADD_phred with DANN_rankscore and Eigen_-raw were determined to be 0.815 and 0.860, respectively, showing that both DANN and Eigen correlate well with CADD. However, interestingly, the distributions of ClinVar and/or Inter-Var P or LP variants were quite different. CADD_phred and DANN_rankscore showed wider distributions in P or LP variants than CADD_phred and Eigen_raw. The Pearson correlation coefficients of CADD_phred with DANN_rankscore and Eigen_raw were 0.127 and 0541, respectively. Interestingly, in both of the scatter plots, *BRCA1* p.L52F, a benign variant annotated as LP by InterVar, showed similar scores to the other P or LP variants in the three parameters. S3 Table shows the details of the computational scoring for the ClinVar/InterVar P or LP variants. The ClinVar P or LP variants clearly showed higher average and minimum scores for CADD_phred and Eigen_raw scores for InterVar P or LP variants, but not for DANN_-rankscore. Based on this observation, we decided to use CADD_phred and Eigen_raw for further filtration of potentially pathogenic mutations.

Minor allele frequencies are also critical parameters for interpreting the clinical impact of germline variants. As expected, both CADD_phred and Eigen_raw show weak positive correlations with the reverse logarithmic minor allele frequencies (Pearson correlation coefficients = 0.172 and 0.161, respectively). The CADD_phred and Eigen_raw scores of the InterVar P or LP variants are similar to those of the ClinVar P or LP variants (Table 2), with the exception of the *BRCA1* L52F variant. Based on these comparisons, we defined computational thresholds for possible pathogenic *BRCA* single-nucleotide variants as follows: CADD_phred$\geq$25.9, Eigen_raw$\geq$0.501, and MAF$\leq$0.0003 (Fig 1A).

Eight *BRCA* variants that fulfill these three criteria defined by the ClinVar P or LP variants are present in 3.5KJPNv2 (Table 2). One of these, *BRCA2* p. G1529R, is annotated as "benign" or "likely benign" by ClinVar and InterVar, respectively. This variant is quite rare but found in two different ethnic groups, namely, African-Americans and non-Finnish Europeans (minor allele frequencies of 0.0003 and 0.0007, respectively; Table 2). Because ClinVar annotated the

**Table 2. Summary of candidate "pathogenic" variants of *BRCA* genes in 3.5KJPN version 2.**

| Class | Position | Ref | Alt | Ref. Gene | ExonicFunc.refGene | AAChange | InterVar | Clinvar | 3.5KJPN | CADD | Eigen | OncoKB | BRCA Exchange |
|---|---|---|---|---|---|---|---|---|---|---|---|---|---|
| InterVar P or LP | 32903605 | TG | - | *BRCA2* | frameshift deletion | p.T219fs | P | P | 0.0001 | . | . | NA | No data |
| | 32911577 | - | T | *BRCA2* | frameshift insertion | p.M1029fs | LP | UNK | 0.0001 | . | . | NA | No data |
| | 32913262 | GT | - | *BRCA2* | frameshift deletion | p.K1590fs | LP | UNK | 0.0001 | . | . | NA | No data |
| | 32914066 | AATT | - | *BRCA2* | frameshift deletion | p.T1858fs | P | P | 0.0003 | . | . | NA | Pathogenic |
| | 32914210 | CT | - | *BRCA2* | frameshift deletion | p.N1906fs | P | P | 0.0001 | . | . | NA | No data |
| | 32914894 | TAACT | - | *BRCA2* | frameshift deletion | p.N2134fs | P | P | 0.0001 | . | . | NA | No data |
| | 32920978 | C | T | *BRCA2* | stopgain | p.R2318X | P | P | 0.0003 | 46 | 0.506 | NA | Pathogenic |
| | 32930651 | G | A | *BRCA2* | nonsynonymous SNV | p.G2508S | LP | Conflicting | 0.0003 | 34 | 0.924 | Likely Neutral | Not reviewed |
| | 32930714 | G | - | *BRCA2* | frameshift deletion | p.G2529fs | LP | UNK | 0.0001 | . | . | NA | No data |
| | 32936719 | A | C | *BRCA2* | nonsynonymous SNV | p.N2622T | LP | UNK | 0.0001 | 28.7 | 0.965 | NA | No data |
| | 32937362 | A | G | *BRCA2* | nonsynonymous SNV | p.I2675V* | LP | P/LP | 0.0001 | 25.9 | 0.756 | Likely Oncogenic | Not reviewed |
| | 32944563 | G | A | *BRCA2* | nonsynonymous SNV | p.A2786T | LP | Conflicting | 0.0001 | 28 | 0.585 | Likely Oncogenic | Not reviewed |
| | 32944612 | C | T | *BRCA2* | nonsynonymous SNV | p.P2802L | LP | UNK | 0.0003 | 32 | 0.291 | NA | Not reviewed |
| | 32954267 | G | A | *BRCA2* | nonsynonymous SNV | p.V3081I | LP | UNK | 0.0001 | 23.8 | -0.348 | NA | Not reviewed |
| | 41197729 | T | C | *BRCA1* | nonsynonymous SNV | p.Y1853C | LP | LP | 0.0003 | 27 | 0.694 | Likely Oncogenic | Not reviewed |
| | 41215947 | C | T | *BRCA1* | nonsynonymous SNV | p.R1699Q | LP | Conflicting | 0.0003 | 35 | 0.871 | Likely Oncogenic | Not reviewed |
| | 41223120 | T | A | *BRCA1* | nonsynonymous SNV | p.Q1604L | LP | UNK | 0.0001 | 18.26 | -0.385 | NA | No data |
| | 41226421 | - | A | *BRCA1* | frameshift insertion | p.V1534fs | LP | UNK | 0.0001 | . | . | NA | No data |
| | 41228562 | T | C | *BRCA1* | nonsynonymous SNV | p.K1476R | LP | UNK | 0.0001 | 23.1 | 0.12 | NA | No data |
| | 41244334 | G | - | *BRCA1* | stopgain | p.L1072X | P | P | 0.0001 | . | . | NA | Pathogenic |
| | 41244748 | G | A | *BRCA1* | stopgain | p.Q934X | P | P | 0.0001 | 35 | 0.501 | NA | Pathogenic |
| | 41258497 | A | T | *BRCA1* | stopgain | p.L63X | P | P | 0.0003 | 39 | 0.807 | NA | Pathogenic |
| Computational +MAF | 32900706 | G | T | *BRCA2* | nonsynonymous SNV | p.S196I | VUS | VUS | 0.0003 | 29.8 | 0.783 | Likely Oncogenic | Not reviewed |
| | 32930669 | A | G | *BRCA2* | nonsynonymous SNV | p.K2514E | VUS | VUS | 0.0001 | 32 | 0.865 | NA | Not reviewed |
| | 32944605 | C | T | *BRCA2* | nonsynonymous SNV | p.P2800S | VUS | VUS | 0.0001 | 33 | 0.864 | NA | Not reviewed |
| | 32968948 | T | G | *BRCA2* | nonsynonymous SNV | p.W3127G | VUS | UNK | 0.0001 | 29.3 | 0.764 | NA | No data |
| | 41197783 | C | T | *BRCA1* | nonsynonymous SNV | p.R1835Q | VUS | VUS | 0.0001 | 34 | 0.738 | Likely Oncogenic | Not reviewed |
| | 41215957 | C | T | *BRCA1* | nonsynonymous SNV | p.V1696M | VUS | VUS | 0.0001 | 33 | 0.754 | Likely Oncogenic | Not reviewed |
| | 41256212 | G | A | *BRCA1* | nonsynonymous SNV | p.S123F | VUS | UNK | 0.0001 | 28.2 | 0.686 | NA | Not reviewed |
| Benign | 32913077 | G | A | *BRCA2* | nonsynonymous SNV | p.G1529R | Benign | Likely benign | 0.0001 | | | NA | Benign / Little Clinical Significance |

\* Known as a splicing error-causing variant [38].

variant as "likely benign," we excluded it from further analysis. A summary of the filtering criteria is shown in Fig 1A.

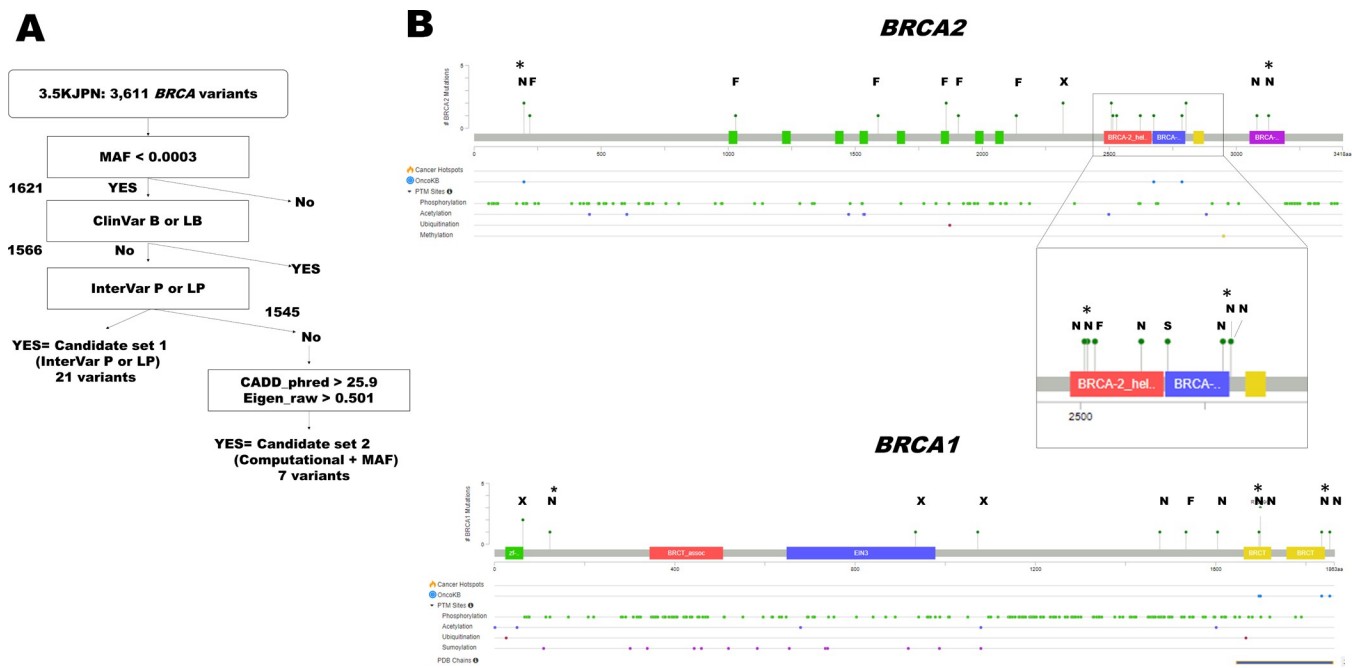

**Fig 1. Relationships between the variant impact scores for the *BRCA* genes in 3.5KJPNv2.** Panel a. Schematic diagram of filtering steps for candidate pathogenic variants in the *BRCA* genes. The details of the filtering process are described in the main text. MAF indicates minor allele frequency. Panel b. Distribution of candidate pathogenic variants of BRCA cDNA in 3.5KJPNv2. Schematic diagram of the BRCA1 and BRCA2 cDNA generated with Mutation Mapper on the cBioPortal. "F," "N," "S," and "X" indicate frameshifts, nonsynonymous single-nucleotide variants, splicing error variants, and stopgains, respectively. The height of lollipops indicates the number of cases found in 3.5KJPNv2. Asterisks indicate variants in the computational + MAF set.

We also tested these criteria for the 134 potentially pathogenic *BRCA* variants for women that were shown to be enriched in breast cancer cases in a previous study [14]. Among them, only 13 variants are found in the latest version of the Japanese whole-genome reference panel (4.7KJPN: jmorp database: https://jmorp.megabank.tohoku.ac.jp/202001/) and all of the available MAFs are $\leq 0.0003$. Eighty-seven variants are annotated as P or LP for both ClinVar and InterVar and 130 variants are annotated as P or LP in either ClinVar or InterVar. Four variants were annotated as "pathogenic" by Momozawa et al. [14], but not annotated as P or LP by InterVar (S4 Table). Among them, three variants showed high CADD_phred (24–35) and Eigen_raw scores (0.571–0.871) (S4 Table). One exception is *BRCA1* p.K1095E, which is annotated as "likely benign" by InterVar and neither the CADD_phred nor the Eigen_raw score reaches our criteria to define it as pathogenic. Therefore, our criteria correspond well to the previous studies.

A summary of the variants identified in this study is shown in Table 2 and the distribution of the candidate pathogenic variants in the BRCA proteins is shown in Fig 1B. The nonsynonymous variants tend to localize at the C-terminal of the genes, while the frameshift indels and stopgains are localized between the N-terminal and the middle of the protein sequence. *BRCA2* I2675V is known as a "splicing error-causing variant" [44] and it is the most C-terminal-end variant causing large structural changes in the BRCA2 mRNA in our collection. So far there is no other splicing error-causing variants of the BRCA1/2 genes in the 3.5KJPN. As shown in Table 1, there are two splicing-affected variants (BRCA2:g.chr13: 32890558AGdel and BRCA2:g.chr13: 32937315G>A) in the gnomAD-EAS, indicating that we did not miss a large numbers of splicing error causing variants in the *BRCA1/2* genes in the 3.5KJPN. We obtained additional annotations at the cBioPortal to draw a schematic diagram; three

candidate pathogenic variants identified based on the three criteria are annotated as likely oncogenic, as well as four InterVar P or LP variants (Table 2). This indicates that our approach can effectively identify the pathogenic variants in the *BRCA* genes. Sugano et al. described *BRCA2* Y1853C as a VUS, although both ClinVar and InterVar annotated it as LP [9]. Later, Kawatsu et al. showed the pathogenic potential of this variant by experimental and genetic analyses [45]. Similarly, another variant, *BRCA2* p.G2508S, is annotated as "likely neutral" by the OncoKB database. However, this variant was recently described as "moderately oncogenic" by Shimelis et al., based on a genome-wide association study of more than 12,000 cases and controls [46]. Therefore, we decided to include this variant for further study.

In the variant call procedures by Tadaka et al., there is no step for ruling out the false positives caused by clonal hematopoiesis [47]. The basic quality control steps when generating 3.5KJPNv2 were described by Tadaka et al. [20], suggesting that some of the *BRCA* variants analyzed in this study might have originated from somatic mutations in the blood leucocytes from the cohort participants. However, we believe that clonal hematopoiesis should not have contributed substantially to our dataset for the following reasons. First, Tadaka et al. used GATK haplotypecaller for variant calling for 3.5KJPNv2, which is suitable to detect variants in near-diploid genomes. Thus, most of the variants caused by clonal hematopoiesis would not reach sufficient variant read depth in a WGS sample. In addition, the average age of individuals from whom the samples in 3.5KJPNv2 were derived was around 56 years old [20]. Clonal hematopoiesis occurs mainly in the elderly, becoming prominent in those aged over 65 [47]. Hence, clonal hematopoiesis may not have strongly affected our results.

## Potentially pathogenic *BRCA* variant carriers tend to have cancer-prone family histories

Members of the TMM CommCohort reported their individual and family histories of various disorders including cancers by completing a paper-based questionnaire. It is possible that the *BRCA* pathogenic variant carriers and their family members would suffer from cancers more often than other cohort members and their family members. Fig 2 indicates the numbers of cases of cancer among the participants themselves, their family members, and their spouses. Although the number of non-carrier participants was more than 1,500 times greater than the number of InterVar P or LP and computational + MAF-selected *BRCA* variant carriers, the overall profiles of cancer onset were similar. For example, fathers of the participants suffered more from cancers than mothers, regardless of the participants' status in terms of *BRCA* variants. A prominent difference between those definitely carrying potentially pathogenic *BRCA* variants and the rest of the cohort was in the rate of cancer-bearing sisters: the InterVar P or LP carriers were shown to have a much higher rate of cancer-bearing sisters than the rest of the cohort (Fig 2 and S5 Table; p = $3.08 \times 10^{-5}$, chi-squared test with Yates' correction). In addition, the rate of cancer-bearing offspring was higher in the InterVar P or LP carriers than in the others with marginally significant (p = 0.041). Interestingly, the rate of cancer onset of the participants themselves did not differ markedly between the InterVar P or LP carriers and the rest of the cohort members. This may be reasonable as nearly half of the InterVar P or LP carriers are male and thus are less likely to suffer from *BRCA*-related breast cancers (S5 Table).

Numbers of positive cases of self-reported individual and family histories of cancer among the TMM CommCohort participants. Vertical axes indicate the number of cases with positivity for each item below the horizontal axis. The right and left axes indicate the *BRCA* candidate pathogenic variant-positive and -negative cases, respectively. The scales of the vertical axes are adjusted by showing the "Family" bars at the same height. "Total cases" indicates the number of cases analyzed, while "Self" indicates individual past history of any malignancy. "Father,"

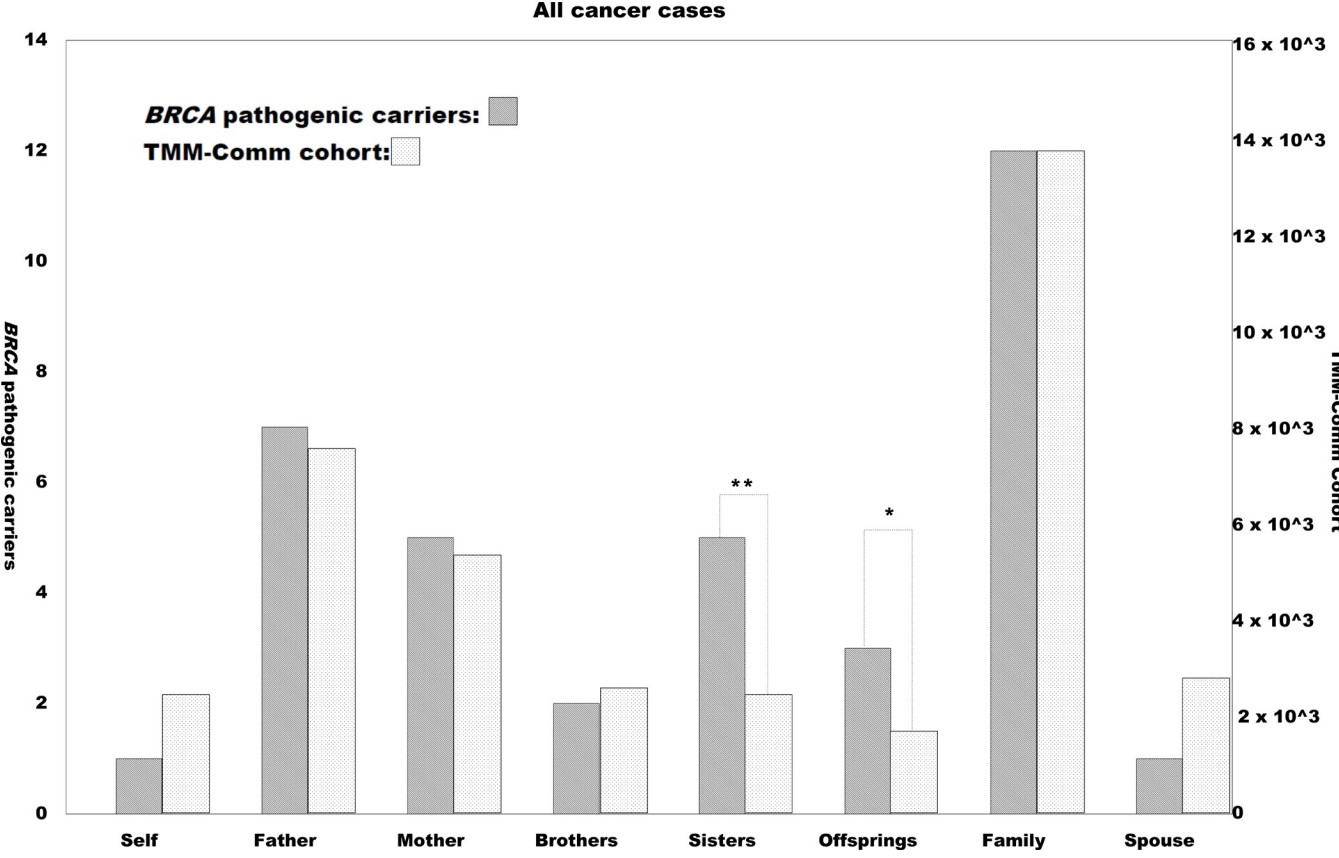

**Fig 2. Preponderance of individual and family histories of cancer for the TMM CommCohort.**

"Mother," "Brother," "Sister," "Offspring," and "Spouse" indicate the cancer-related histories of the participants' family members. "Family" indicates a case of any cancer among any of the blood relatives, except the participants themselves. Cases in which the "Spouse" was positive are not included in "Family." Solid and gray bars represent numbers of cases positive for the *BRCA* candidate pathogenic variants and the rest of the TMM CommCohort cases, respectively. Asterisks indicate statistically significant differences (single: $p < 0.05$, double: $p < 10^{-4}$) upon comparison with the total analyzed TMM CommCohort cases (Fig 2).

Recent progress in bioinformatics may open up a completely different path for filtering the VUSs in hereditary disorders, namely, artificial intelligence-mediated approaches. One example of this is CADD, which was reported in 2014 [31]. CADD scores are based on calculations of all of the possible 84 billion single-nucleotide changes in the human genome. Such calculation is based on machine learning using the evolutionarily conserved "proxy-neutral" variants found in both apes and humans and the recently emerged "proxy-pathogenic" rare variants in the human genome alone [48]. In 2016, a further dataset, Eigen, was released, for which calculation was performed without training data but with a principal component that gives the largest diversity among the variants prepared from all possible single-nucleotide changes in the human genome [33]. These annotation tools have achieved some clinically significant findings in genome-wide association studies (for example, see [49, 50]). Recently, The ICGC/TCGA Pan-Cancer Analysis of Whole Genomes Consortium successfully applied CADD to estimate the biological impacts of cancer mutations [51]. Therefore, computational scoring is suggested to be very powerful at predicting the clinical impact of single-nucleotide variants in cancer-

predisposing genes. The present study shows the potential for applying this approach to find pathogenic variations in cancer-predisposing syndromes by using genome reference panels with precise MAF estimation. This study shows that the MAF estimate for the general population is much more useful for the annotation of pathogenic variants than the biased collection of population samples.

Recently, Findlay et al. reported that a saturation genomics-based approach could functionally characterize more than 4,000 *BRCA1* variants that are in the functionally critical regions [52]. Thirteen *BRCA1* variants in the 3.5KJPNv2 corresponds to the list of Findlay et al. and among them, three loss of function variants were annotated by Findlay et al. (L63X, R71S, and Y1853C). Specifically, among them, there were two discordant variants between the work of Findlay et al. and the present study. *BRCA1* p.R71S was not picked up by our survey but annotated as "loss of function" in the dataset by Findlay et al., while *BRCA1* V1696M was picked up by our survey but annotated as "functional" in the same dataset. It is possible that the pathogenicity of *BRCA* variants would be affected by other genetic modifiers and/or environmental factors. Most of the computational methods for estimating the impact of genetic variants depend on "known datasets" when they perform machine learning. There are probably many "unknown factors" that are essential for the correct estimation of pathogenicity of variants. Further studies should be performed to provide new and critical information for the computational estimation of pathogenicity of genetic variations. Follow-up of the carriers of these variants in prospective cohort studies may also provide clues to resolving any discordant results.

There was a significant preponderance of cancer in the family histories of those with potentially pathogenic *BRCA* variants only among the sisters of TMM CommCohort members. The carriers found in the TMM CommCohort were mainly male and the female carriers were relatively young, so they themselves had not yet accumulated many cancer cases. A preponderance of a history of cancer in the mothers was not observed, but the mothers should have been aged over 80, so their accumulation of sporadic cancers would have obscured the HBOC cases. Our study suggests that the self-reported data of the TMM CommCohort are useful to analyze the genotype–phenotype relationships, at least in cancer-predisposing syndromes.

It is not easy to estimate the clinical significance of the VUSs that may have clinically significant effects on the hosts' predisposition for cancer, but with relatively low penetrance. It is an important insight that VUSs may have moderate but significant effects on cancer onset that can be reduced by personalized health care based on information on the genetic variant. Around 10 years ago, a review paper by Berger et al. proposed that haploinsufficiency is not so uncommon in the onset of cancer in HBOC patients with pathogenic variants in the *BRCA* genes [53]. Moderately deleterious variants are also critical for the successful establishment of precision medicine and/or personalized health care [54]. The carriers in moderately penetrant HBOC families may not be critical to prompt radical interventions such as prophylactic surgery, but the carriers may be encouraged to continue undergoing close health checks to detect HBOC cancers as early as possible.

## Conclusions

The present study indicates that a large dataset of Japanese whole-genome sequencing data (3.5KJPNv2) includes definitely and potentially pathogenic variants in representative genes responsible for HBOC: *BRCA1* and *BRCA2*. ClinVar and the ACMG-guided annotation tool InterVar detected more than 20 variants as pathogenic or likely pathogenic, including one obviously benign variant in 3.5KJPNv2. In addition, the use of the combination of computational scoring and MAF picked up another eight candidates, including one likely benign mutant as defined by ClinVar. Some of the variants show concordance with other databases in

terms of the pathogenic annotations. The self-reported individual and family histories of the carriers of potentially pathogenic *BRCA* variants were analyzed and the carriers' sisters showed a significant history of cancer themselves. This study indicates that prospective genomic cohort studies are a powerful tool for identifying pathogenic variants. The present study should be useful for identifying such moderately deleterious variations in populations and contribute to the development of personalized health care based on individual genomic information.

## Supporting information

**S1 Table. Functional annotations of the *BRCA* gene variants in 3.5KJPNv2 and gnomAD.**
(XLSX)

**S2 Table. InterVar P or LP variants in the *BRCA* genes of the RIKEN 2,234 Japanese whole-genome sequence dataset.**
(XLSX)

**S3 Table. Comparison of scores for pathogenic variants in the *BRCA* genes in 3.5KJPN.**
(XLSX)

**S4 Table. Details of "pathogenic" *BRCA* variants but not P or LP by InterVar in the paper by Momozawa et al.**
(XLSX)

**S5 Table. Statistics of the sisters or offspring cancer histories of candidate BRCA pathogenic variants carriers in the TMM-Comm cohort.**
(XLSX)

## Acknowledgments

We thank all past and present members of Tohoku Medical Megabank Organization at Tohoku University (present members are listed at https://www.megabank.tohoku.ac.jp/english/a191201/). We also thank Edanz Group (https://en-author-services.edanzgroup.com/ac) for editing the English text of a draft of this manuscript.

## Author Contributions

**Conceptualization:** Hideki Tokunaga, Soichi Ogishima, Yoh Watanabe, Shogo Shigeta, Yumi Yamaguchi-Kabata, Nobuo Fuse, Nobuo Yaegashi, Jun Yasuda.

**Data curation:** Hideki Tokunaga, Keita Iida, Atsushi Hozawa, Soichi Ogishima, Yumi Yamaguchi-Kabata, Shin Ito, Kazuki Kumada, Jun Yasuda.

**Formal analysis:** Keita Iida, Atsushi Hozawa, Shin Ito.

**Funding acquisition:** Hideki Tokunaga, Muneaki Shimada, Masayuki Yamamoto, Nobuo Yaegashi, Jun Yasuda.

**Investigation:** Hideki Tokunaga, Keita Iida, Jun Yasuda.

**Methodology:** Jun Yasuda.

**Project administration:** Hideki Tokunaga, Jun Yasuda.

**Resources:** Atsushi Hozawa, Shu Tadaka, Fumiki Katsuoka, Kengo Kinoshita.

**Software:** Kengo Kinoshita, Jun Yasuda.

**Supervision:** Hideki Tokunaga, Atsushi Hozawa, Muneaki Shimada, Yohei Hamanaka, Nobuo Fuse, Kengo Kinoshita, Masayuki Yamamoto, Nobuo Yaegashi.

**Writing – original draft:** Hideki Tokunaga, Jun Yasuda.

**Writing – review & editing:** Hideki Tokunaga, Yumi Yamaguchi-Kabata, Nobuo Yaegashi, Jun Yasuda.

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
