## [Decision Letter · Decision Letter 0]

23 Sep 2020

PONE-D-20-21851

Novel candidates of pathogenic variants of the *BRCA1* and *BRCA2* genes in a 3,552 Japanese whole-genome sequence dataset (3.5KJPNv2)

PLOS ONE

Dear Dr. Yasuda,

Thank you for submitting your manuscript to PLOS ONE. After careful consideration, we feel that it has merit but does not fully meet PLOS ONE’s publication criteria as it currently stands. Therefore, we invite you to submit a revised version of the manuscript that addresses the points raised during the review process.

We look forward to receiving your revised manuscript.

Kind regards,

Yonglan Zheng

Academic Editor

PLOS ONE

3. To comply with PLOS ONE submission guidelines, in your Methods section, please provide additional information regarding your statistical analyses. For more information on PLOS ONE's expectations for statistical reporting, please see https://journals.plos.org/plosone/s/submission-guidelines.#loc-statistical-reporting.

4.We note that you have indicated that data from this study are available upon request. PLOS only allows data to be available upon request if there are legal or ethical restrictions on sharing data publicly. For information on unacceptable data access restrictions, please see http://journals.plos.org/plosone/s/data-availability#loc-unacceptable-data-access-restrictions.

Reviewers' comments:

Reviewer's Responses to Questions

**Comments to the Author**

1. Is the manuscript technically sound, and do the data support the conclusions?

Reviewer #1: Partly

Reviewer #2: No

2. Has the statistical analysis been performed appropriately and rigorously? 

Reviewer #1: Yes

Reviewer #2: Yes

3. Have the authors made all data underlying the findings in their manuscript fully available?

Reviewer #1: Yes

Reviewer #2: Yes

4. Is the manuscript presented in an intelligible fashion and written in standard English?

Reviewer #1: Yes

Reviewer #2: Yes

5. Review Comments to the Author

Reviewer #1: Overall:

This study reviews identifies participants in a whole-genome databank with BRCA1/2 mutations that can be considered pathogenic. They identify several variants that are reportedly novel based on ClinVar annotation and discuss the validity of these variants regarding possible pathogenicity.

Strengths:

-The introduction does contain a thorough review of prior studies regarding BRCA1 and BRCA2 mutations in the Japanese population

-Computational annotation of genomes is clearly described using standard resources in the field

-The paper highlights several methods that are used to validate pathogenicity of BRCA1/2 mutations, and the self-reported findings of family members with cancer and variant annotations is an interesting one highlighted in this paper.

Areas for Improvement:

Major:

-The authors do not describe the work of international consortia in this area such as the ENIGMA consortium (https://enigmaconsortium.org/library/enigma-publications/). When they describe the work of prior authors, they refer to other studies in the Japanese population (which is thoroughly reviewed), but not other international consortia or databases which have been dedicated to BRCA pathogenicity. This is of concern as it is not clear if the authors searched other sources to determine that their findings were novel.

-The authors do not clarify the aim of their approach and its novelty relative to prior studies in the Japanese population with larger patient samples. This is a large sample of whole genomes and thereby offers novelty to identify pathogenic mutations that may have not have been previously identified through exome means. It is instead unclear if they are interested in ascertaining the general population frequency of pathogenic mutations in the Japanese population, or in validation of their computational strategies. The authors need to clearly define their aim in the abstract and introduction.

-From their methods, it is not clear if and how the authors address splicing variants and noncoding variants if their goal was to identify potentially pathogenic variants. Intervar does not address these – it uses exonic variants only. These would not be consistently reported in ClinVar, and it is known that several splicing variants in BRCA1 and BRCA2 can lead to clinical pathogenic findings. This would leverage the major benefit offered by the whole genome data.

-From their methods, it is not clear how the authors can confirm that their findings are specifically germline and not due to clonal hematopoiesis. Other approaches to identifying pathogenic germline variants do include methods of quality control in this area for confirmation or allude to this issue in the discussion if this cannot be resolved with the methods used.

-While comparison to GnomAD is standard practice, this is typically done in the context of population-specific evaluations of specific allele frequencies. The conclusions regarding BRCA1/2 variants across both databases is not a typical use of GnomAD nor a clear conclusion with all population data in GnomAD aggregated. GnomAD should be used to discuss specific mutations identified or specific populations. (Would recommend that Table 1 be revised accordingly.)

-Please revise or provide a table to clarify the population studied in the database and range of number of mutations and types of mutations identified per individual (not average number per individual, which is difficult to interpret). This information is not clear from the paper as written. Additionally, the utility of the data provided in the Figures and Tables is frankly mixed, with some information very helpful for the reader and some information not clearly helpful towards the authors’ conclusions. To this end: Figure 1C is appropriate to serve as Figure 1. Figure 1 A and B should either be provided later in the text (with that section moved down) or added to supplementary figures. Figure 2 should be moved up (and could also be a candidate for Figure 1). Please move supplementary Table 2 to the actual paper or provide a table of the novel variants identified by the authors and the annotation information as they provide in the text. Please move Table 2 to the supplement as it is not clear that this provides extensive additional data. Supplementary Table 4 should be moved to the main text.

-Explanations or interpretations from the authors in the results outside of description of the actual results should be moved to the methods or discussion sections as appropriate. The results section is extremely long as a result and the discussion section is too short.

*Regarding the reliability of short-read sequence data

*Findings as interpreted by the authors regarding GnomAD

*CADD vs. Eigen scores

*Data regarding specific variants from outside sources

Minor:

-The title was slightly unclear: Would rephrase to

Novel candidates of pathogenic variants of the BRCA1 and BRCA2 genes from a dataset of 3,552 Japanese whole genomes (3.5KJPNv2)

-The introduction contains conflations regarding methods of identifying pathogenic mutations. For example, the reference to allelic dropout in the paper by Yost et al. is describing germline mutations taken from patients’ tumors, which, while a means of confirming pathogenicity, is subject to its own issues. The question that the authors are asking, though, is regarding unaffected carriers who have BRCA1 and BRCA2 mutations in this cohort. It is confusing to switch back and forth between germline testing by blood / via unaffected carriers and via tumor in affected carriers given the significant differences between these two methods unless this is delineated clearly. The introduction and references should be reviewed to clarify prior methods of identifying BRCA1/BRCA2 pathogenic variants and associated literature, and then also to discuss the findings that have been specific to the Japanese population.

-For the methods, any ANNOVAR/annotation software using the ClinVar database should have the date of reference noted, since ClinVar is updated regularly.

-It would be very helpful for the authors to review the specific methods from prior work that are relevant for their study. For example, an extremely brief review regarding the criteria for WGS selection in the Megabank, depth of sequencing, as well as the methods of how these sequences are obtained (e.g. from whole blood?) and how families may be linked in the Project data.

-For the methods, it would be helpful for there to be quantitative descriptions of the filtering process and use of the self-questionnaire data, as this is not replicable based on the current description.

-Computational estimation of pathogenicity is a data source, but this is an ongoing point of information used in interpreting pathogenicity. The authors’ conclusion at one point between conflicting data sources that “pathogenicity of BRCA variants would be affected by other genetic modifiers and/or environmental factors,” while absolutely true, is not as applicable in discussing the discordance between different methods of estimating pathogenicity (computer vs. saturation genomics modeling). Rather, the question is regarding the fallibility of these estimation approaches. Given the authors’ findings regarding some “pathogenic” mutations annotated as such but clearly benign on futher review, this warrants a significant component of the discussion.

-The comparison between ClinVar and InterVar mutations in the tables is unclear. Does this mean “known pathogenic” and “annotated as pathogenic and novel, but under review”?

-Formatting of captions for tables and figures is not consistent.

-The final paragraph of the results is written as though to conflate variants of uncertain significance and moderate penetrance. Please revise this.

-Regarding data access: It would be more appropriate for the authors to state that they do not own the data themselves, but access to it is governed by the steering committee. dbGAP in the US is available but under the same restrictions, and patients’ privacy is honored.

Reviewer #2: The authors indicates that a large dataset of Japanese whole-genome sequencing data includes pathogenic variations in BRCA1/2 genes responsible for HBOC. ClinVar and InterVar detected more than 20 variants as pathogenic or likely pathogenic. The use of the combination of computational scoring and MAF picked up another eight candidates, including one likely benign mutant as defined by ClinVar. The self-reported individual and family histories of the carriers of potentially pathogenic BRCA variants were analyzed and the carriers’ sisters showed a significant history of cancer themselves.

There are major comments on this study.

1. They use ClinVar, InterVar, computational scoring systems and MAF to evaluate the pathogenicity of the BRCA variants found in their cohort. The approaches are all common and novelty of the study is limited.

2. It is not clear why the difference was observed only in the cancer-bearing sisters of the TMM CommCohort. It seems that the paper-based questionnaires is not so robust to differentiate the pathogenic BRCA variant carriers or to evaluate BRCA annotation systems.

3. Functional analysis is recommended to confirm their annotation is accurate for the variants discordance was observed between ClinVar and their annotation system.

Minor comments are the following:

1. The meanings of sentence p20, l297-299 is not clear.

2. p21, l309 cBioPortal.

3. The total number of all candidate in Figure 3 should be 27 considering male and female number.

4. Poor figure resolution.

6. PLOS authors have the option to publish the peer review history of their article (what does this mean?). If published, this will include your full peer review and any attached files.

Reviewer #1: No

Reviewer #2: No

---

## [Author Response · Author response to Decision Letter 0]

20 Oct 2020

PONE-D-20-21851

Novel candidates of pathogenic variants of the BRCA1 and BRCA2 genes in a 3,552 Japanese whole-genome sequence dataset (3.5KJPNv2)

PLOS ONE

We believe that the revised manuscript meets the requirements of PLOS ONE.

We removed the following two sentences that imply the use of inaccessible data. We believe that this deletion will not affect the main message of our manuscript: 

 We also checked the status regarding smoking and alcohol consumption in the potential HBOC carriers and others in the TMM CommCohort participants; we did not observe any significant difference between them (data not shown).

Intriguingly, there are 0 and 8 overlaps between two datasets for ClinVar and InterVar P or LP variants, respectively (data not shown). 

3. To comply with PLOS ONE submission guidelines, in your Methods section, please provide additional information regarding your statistical analyses. For more information on PLOS ONE's expectations for statistical reporting, please see https://journals.plos.org/plosone/s/submission-guidelines.#loc-statistical-reporting.

We added a “Statistics” subsection in the “Methods” section as follows:

“Statistics

 To analyze the correlations among the three computational estimates of the impacts of variants, we employed R 3.6.1 for calculating the Pearson correlation coefficient. We applied Fisher’s exact test chi-squared test with Yates’ correction for calculating the p-values of the differences in numbers of cancer-bearing family members. 

In terms of access to data from the TMM prospective cohort project, users should obtain approval from the sample and data access committee of the TMM Biobank (Minegishi, 2019). The sample and data access committee consists of experts both inside and outside the TMM. Upon applying to this committee, the Group of Materials and Information Management in the TMM at Tohoku University supports the procedures for data transfer.

In terms of access to the data used in our manuscript, the Group of Materials and Information Management in the Tohoku Medical Megabank Organization at Tohoku University supports the procedures for data transfer. The Group of Materials and Information Management can be contacted at dist@megabank.tohoku.ac.jp.

There are restrictions on the data usage, so we cannot upload our data for replication by a third party without conditions.

We moved “Ethics approval and consent to participate” to the “Methods” section.

Reviewers' comments:

Comments to the Author

5. Review Comments to the Author

Reviewer #1: Overall:

This study reviews identifies participants in a whole-genome databank with BRCA1/2 mutations that can be considered pathogenic. They identify several variants that are reportedly novel based on ClinVar annotation and discuss the validity of these variants regarding possible pathogenicity.

Strengths:

-The introduction does contain a thorough review of prior studies regarding BRCA1 and BRCA2 mutations in the Japanese population

-Computational annotation of genomes is clearly described using standard resources in the field

-The paper highlights several methods that are used to validate pathogenicity of BRCA1/2 mutations, and the self-reported findings of family members with cancer and variant annotations is an interesting one highlighted in this paper.

Areas for Improvement:

Major:

-The authors do not describe the work of international consortia in this area such as the ENIGMA consortium (https://enigmaconsortium.org/library/enigma-publications/). When they describe the work of prior authors, they refer to other studies in the Japanese population (which is thoroughly reviewed), but not other international consortia or databases which have been dedicated to BRCA pathogenicity. This is of concern as it is not clear if the authors searched other sources to determine that their findings were novel.

Thank you for your constructive comment. We added the following text in the “Introduction” section:

“Moreover, the BRCA Challenge project established an open access database, BRCA Exchange for providing reliable and easily accessible variant data for better clinical treatments of HBOC [12]. As of October 2020, the BRCA Exchange database has collected more than 40,000 variants in the BRCA1/2 genes from major clinical databases and estimated their pathogenicity under expert peer review in collaboration with the ENIGMA consortium [13]. The purposes of this comprehensive database are to provide reliable and easily accessible variant data interpreted for the high-penetrance phenotype of HBOC and to develop a model database for the utilization and sharing of public data to provide better clinical treatments of hereditary disease. In this database, there are more than 4,900 variants annotated as “pathogenic” by the ENIGMA consortium. “.

-The authors do not clarify the aim of their approach and its novelty relative to prior studies in the Japanese population with larger patient samples. This is a large sample of whole genomes and thereby offers novelty to identify pathogenic mutations that may have not have been previously identified through exome means. It is instead unclear if they are interested in ascertaining the general population frequency of pathogenic mutations in the Japanese population, or in validation of their computational strategies. The authors need to clearly define their aim in the abstract and introduction.

 Thank you very much for the constructive comment. Both of the points that the reviewer raised are of interest to us. One interest is ascertaining the precise frequency of pathogenic mutations in the general Japanese population. The other is validating computational strategies to identify moderately pathogenic variants. Therefore, we amended our manuscript in the “Abstract” and “Introduction” sections to reflect this.

In the “Abstract” section

“Identification of frequencies of definitely pathogenic germline variants in a population in two major hereditary breast and ovarian cancer syndrome (HBOC) genes, BRCA1/2, will be essential to estimate the number of HBOC patients. In addition, moderately pathogenic HBOC gene variants that contribute to increase the risk of breast and ovarian cancers in a population and identification of such variants will be critical in the establishment of personalized healthcare. The prospective cohort with genome analyses will provide both information”.

In the “Introduction” section

“Patient-based studies for identifying germline pathogenic variants are very effective for identifying potential pathogenic variants, but cannot estimate the frequencies of those alleles in the general population, which will be critical to estimate the number of HBOC patients in a community. In addition, moderately pathogenic HBOC gene variants will contribute to increase the risk of breast and ovarian cancers in a population and identification of such variants will be critical in the establishment of personalized healthcare. The carriers of moderately pathogenic HBOC variants will not have undergone the drastic prophylactic modalities but frequent examination will be recommendable for earlier detection of the cancers. The prospective cohort with genome analyses will provide both information”. 

-From their methods, it is not clear if and how the authors address splicing variants and noncoding variants if their goal was to identify potentially pathogenic variants. Intervar does not address these – it uses exonic variants only. These would not be consistently reported in ClinVar, and it is known that several splicing variants in BRCA1 and BRCA2 can lead to clinical pathogenic findings. This would leverage the major benefit offered by the whole genome data.

Thank you for the critical comment. We completely agree with the reviewer’s point that the alteration of splicing patterns by genetic variation is critical and whole-genome sequencing data would outperform the exome data for the detection of unknown pathologic splicing variants. In the manuscript by Li et al., the authors point out that InterVar software can infer splicing impacts using ANNOVAR from the dbscsnv11 database. Therefore, the splicing variants that can be detected by dbscsnv11 will be detected by InterVar, although the original paper for dbscsnv11 discussed the intrinsic limitations of their software. Therefore, we added the following text to the “Introduction”:

“The main benefit of the whole-genome sequencing of the dataset is that it provides more comprehensive information of the structure of the two HBOC genes than the exome-based approach”.

-From their methods, it is not clear how the authors can confirm that their findings are specifically germline and not due to clonal hematopoiesis. Other approaches to identifying pathogenic germline variants do include methods of quality control in this area for confirmation or allude to this issue in the discussion if this cannot be resolved with the methods used.

Thank you for the critical comments. The reviewer’s point is correct that there is no step for ruling out the false positives caused by clonal hematopoiesis. The basic quality control steps when generating 3.5KJPNv2 were described by Tadaka et al. in Human Genome Variation (2019) 6:28. There is no description of how to avoid clonal hematopoiesis in the manuscript. However, we believe that clonal hematopoiesis should not have contributed substantially to our dataset for the following reasons. 

First, Tadaka et al. used GATK haplotypecaller for variant calling, which is suitable to detect variants in near-diploid genomes. Thus, most of the variants caused by clonal hematopoiesis would not reach sufficient variant read depth in a WGS sample. Second, the average age of the individuals from whom the samples in 3.5KJPNv2 were derived was around 56 years, as estimated by Tadaka et al. in Human Genome Variation (2019) 6:28. However, clonal hematopoiesis mainly occurs in the elderly, with it being reported to become prominent in Icelandic individuals older than 65 (Zink et al., Blood 2017). Thus, as the reviewer suggested, we added a discussion about clonal hematopoiesis in the “Results and Discussion” section.

“In the variant call procedures by Tadaka et al., there is no step for ruling out the false positives caused by clonal hematopoiesis [47]. The basic quality control steps when generating 3.5KJPNv2 were described by Tadaka et al. [20], suggesting that some of the BRCA variants analyzed in this study might have originated from somatic mutations in the blood leucocytes from the cohort participants. However, we believe that clonal hematopoiesis should not have contributed substantially to our dataset for the following reasons. First, Tadaka et al. used GATK haplotypecaller for variant calling for 3.5KJPNv2, which is suitable to detect variants in near-diploid genomes. Thus, most of the variants caused by clonal hematopoiesis would not reach sufficient variant read depth in a WGS sample. In addition, the average age of individuals from whom the samples in 3.5KJPNv2 were derived was around 56 years old [20]. Clonal hematopoiesis occurs mainly in the elderly, becoming prominent in those aged over 65 [47]. Hence, clonal hematopoiesis may not have strongly affected our results.”

-While comparison to GnomAD is standard practice, this is typically done in the context of population-specific evaluations of specific allele frequencies. The conclusions regarding BRCA1/2 variants across both databases is not a typical use of GnomAD nor a clear conclusion with all population data in GnomAD aggregated. GnomAD should be used to discuss specific mutations identified or specific populations. (Would recommend that Table 1 be revised accordingly.)

Thank you very much for your constructive comment. We agree with the points raised and amended the structure of the manuscript accordingly. We rearranged previous Table 1 as Supplementary Table 1 and renumbered the other supplementary tables accordingly (see also the following point). We then changed the main text as follows:

“To investigate the population specificity, we extracted ClinVar and InterVar P or LP variants found in East Asian populations in the GnomAD database (GnomAD-EAS; Table 1). Intriguingly, there were only four overlaps between 3.5KJPN and GnomAD-EAS for ClinVar and InterVar P or LP variants (Table 1). For example, one of the most prominent BRCA1 pathogenic variants, L63X [9, 10], does not appear in GnomAD-EAS (Table 1). In contrast, the two most prevalent P or LP variants, BRCA2 p.G2508S and p.A2786T, are present in 3.5KJPNv2. These two variants may be commonly distributed among East Asian populations.”.

Amended Table 1 is a large one, so we could not include the new one into this document. Please check it in the revised main text. 

-Please revise or provide a table to clarify the population studied in the database and range of number of mutations and types of mutations identified per individual (not average number per individual, which is difficult to interpret). This information is not clear from the paper as written. Additionally, the utility of the data provided in the Figures and Tables is frankly mixed, with some information very helpful for the reader and some information not clearly helpful towards the authors’ conclusions. To this end: Figure 1C is appropriate to serve as Figure 1. Figure 1 A and B should either be provided later in the text (with that section moved down) or added to supplementary figures. Figure 2 should be moved up (and could also be a candidate for Figure 1). Please move supplementary Table 2 to the actual paper or provide a table of the novel variants identified by the authors and the annotation information as they provide in the text. Please move Table 2 to the supplement as it is not clear that this provides extensive additional data. Supplementary Table 4 should be moved to the main text.

Thank you for your constructive comment. We followed the reviewer’s suggestions. Previous Figures 1C and 2 were changed to Figure 1A and 1B, respectively. We also rearranged Table 2 as Supplementary Table 2. Then, we described the findings appearing in previous Figure 1A and B in the main text as follows:

“The Pearson correlation coefficients of CADD_phred with DANN_rankscore and Eigen_raw are 0.815 and 0.860, respectively, showing that both DANN and Eigen correlate well with CADD. However, interestingly, the distributions of ClinVar and/or InterVar P or LP variants are quite different. CADD_phred and DANN_rankscore show wider distributions in P or LP variants compared with CADD_phred and Eigen_raw. The Pearson correlation coefficients of CADD_phred with DANN_rankscore and Eigen_raw are 0.127 and 0541, respectively. Interestingly, in both of the scatter plots, BRCA1 p.L52F, a benign variant annotated as LP by InterVar, shows at a similar scores to the other P or LP variants in the three parameters. Supplementary Table 3 shows the details of the computational scoring in the ClinVar/InterVar P or LP variants.”

We also removed previous Supplementary Table 4 and updated the main text.

“In addition, the difference of the ratio of cancer-bearing offspring was higher in the InterVar P or LP carriers than in the others, although this was not statistically significant (p = 0.041).” 

The other proposals, such as converting previous Supplementary Table 2 to new Table 2 with all of the described annotations, have also been acted upon. Therefore, the numbering of the figures and tables has been updated throughout the manuscript.

-Explanations or interpretations from the authors in the results outside of description of the actual results should be moved to the methods or discussion sections as appropriate. The results section is extremely long as a result and the discussion section is too short.

*Regarding the reliability of short-read sequence data

*Findings as interpreted by the authors regarding GnomAD

*CADD vs. Eigen scores

*Data regarding specific variants from outside sources

Thank you for the constructive comment. We actually followed the recommendations of the journal itself and used the “Results and Discussion” format that merges the two sections into one. This enabled the “Conclusions” section to be more concise. The reviewer’s points are all important, but some of the issues might be outside the remit of our current study. For example, our data were not focused on the reliability of short read sequences. Similarly, we do not have enough data to discuss which of CADD or Eigen is better. In terms of the GnomAD data, we amended Table 1 and added some discussion as per the reviewer’s suggestion. 

Minor:

-The title was slightly unclear: Would rephrase to

Novel candidates of pathogenic variants of the BRCA1 and BRCA2 genes from a dataset of 3,552 Japanese whole genomes (3.5KJPNv2)

Thank you for this constructive comment. We agree with this and rephrased our manuscript’s title as suggested.

-The introduction contains conflations regarding methods of identifying pathogenic mutations. For example, the reference to allelic dropout in the paper by Yost et al. is describing germline mutations taken from patients’ tumors, which, while a means of confirming pathogenicity, is subject to its own issues. The question that the authors are asking, though, is regarding unaffected carriers who have BRCA1 and BRCA2 mutations in this cohort. It is confusing to switch back and forth between germline testing by blood / via unaffected carriers and via tumor in affected carriers given the significant differences between these two methods unless this is delineated clearly. The introduction and references should be reviewed to clarify prior methods of identifying BRCA1/BRCA2 pathogenic variants and associated literature, and then also to discuss the findings that have been specific to the Japanese population.

Thank you for the critical comment, with which we completely agree. We removed the sentences related to the study by Yost et al. accordingly.

-For the methods, any ANNOVAR/annotation software using the ClinVar database should have the date of reference noted, since ClinVar is updated regularly.

Thank you for the constructive comment. We added information on the ClinVar version (version from December 1, 2015) in the “Methods” section.

-It would be very helpful for the authors to review the specific methods from prior work that are relevant for their study. For example, an extremely brief review regarding the criteria for WGS selection in the Megabank, depth of sequencing, as well as the methods of how these sequences are obtained (e.g. from whole blood?) and how families may be linked in the Project data.

Thank you for the constructive comment. We added a brief review about the TMM project data management in the “Methods” section as follows.

“The whole-genome sequences of some of the participants have been obtained; the criteria for selecting WGS samples are described elsewhere [19, 22]. In brief, the samples for development of the Japanese whole-genome sequencing dataset were selected based on the SNP array data of the samples. Only one sample was picked up from a kinship group to obtain the precise allele frequencies. The whole-genome sequencing was performed with HiSeq 2500 sequencers (Illumina, Inc., San Diego, CA) with a PCR-free protocol from the genomic DNA extracted from whole blood.”.

-For the methods, it would be helpful for there to be quantitative descriptions of the filtering process and use of the self-questionnaire data, as this is not replicable based on the current description.

Thank you for the critical comment. We added the following explanations of filtering the self-questionnaire data to the “Methods” section.

“The self-reported questionnaire data were filtered out for the participants who checked more than 50 items for past and family histories of malignant neoplasms. Most of the participants who checked more than 50 items showed contradictory histories, such as a self-history of ovarian cancer being recorded by male participants. Therefore, we decided to remove such records and obtained 35,136 records as a result. In the statistical analysis comparing carriers of candidate BRCA pathogenic variants and other TMM CommCohort participants regarding self-reported individual and family histories, we employed the binomial distribution to calculate the p-value. Then, we calculated the accumulation of past and family histories only for the items of malignant neoplasms. The questionnaire just asked about the presence or absence of such histories, which could be represented as “0” or “1” for each item. This made it impossible to give a weight to the numbers of affected siblings or offspring.”. 

-Computational estimation of pathogenicity is a data source, but this is an ongoing point of information used in interpreting pathogenicity. The authors’ conclusion at one point between conflicting data sources that “pathogenicity of BRCA variants would be affected by other genetic modifiers and/or environmental factors,” while absolutely true, is not as applicable in discussing the discordance between different methods of estimating pathogenicity (computer vs. saturation genomics modeling). Rather, the question is regarding the fallibility of these estimation approaches. Given the authors’ findings regarding some “pathogenic” mutations annotated as such but clearly benign on further review, this warrants a significant component of the discussion.

 Thank you for the thoughtful comment, with which we completely agree. We added the following text in the “Results and discussion” section accordingly:

“Most of the computational estimation methods for the impact of genetic variations depend on the ‘known data sets’ when they perform machine learning. Probably, there are many ‘unknown factors’ that are essential for correct estimation of variants. Further studies should be necessary to provide the new and critical information for the computational estimation of genetic variations. Follow-up of the carriers of these variants in prospective cohort studies may provide the clue to solve the discordance, too”.

-The comparison between ClinVar and InterVar mutations in the tables is unclear. Does this mean “known pathogenic” and “annotated as pathogenic and novel, but under review”?

Thank you for this comment. The reviewer’s understanding is correct.

-Formatting of captions for tables and figures is not consistent.

Thank you for this comment. There were several inconsistent descriptions in the legends of previous Figure 1a and b. We removed them from the revised version of the manuscript and fixed some symbols in the new Figure 1b. We also revised the tables. We believe that there are now no inconsistencies. 

-The final paragraph of the results is written as though to conflate variants of uncertain significance and moderate penetrance. Please revise this.

Thank you for this thoughtful comment. Accordingly, we amended the text as follows:

“Among those genes, it is not easy to estimate the clinical significance of the VUSs that may have clinically significant effects on the hosts’ predisposition for cancer, but with relatively low penetrance.”.

-Regarding data access: It would be more appropriate for the authors to state that they do not own the data themselves, but access to it is governed by the steering committee. dbGAP in the US is available but under the same restrictions, and patients’ privacy is honored.

Thank you for this constructive comment. In accordance with this, we added a paragraph in the methods section.

“In terms of the access to data from the TMM prospective cohort project, users should obtain approval from the sample and data access committee of the TMM Biobank [17]. This committee consists of experts both inside and outside the TMM. Upon the receipt of an application to the committee, the Group of Materials and Information Management in the TMM at Tohoku University supports the procedures for data utilization”.

Reviewer #2: The authors indicates that a large dataset of Japanese whole-genome sequencing data includes pathogenic variations in BRCA1/2 genes responsible for HBOC. ClinVar and InterVar detected more than 20 variants as pathogenic or likely pathogenic. The use of the combination of computational scoring and MAF picked up another eight candidates, including one likely benign mutant as defined by ClinVar. The self-reported individual and family histories of the carriers of potentially pathogenic BRCA variants were analyzed and the carriers’ sisters showed a significant history of cancer themselves.

There are major comments on this study.

1. They use ClinVar, InterVar, computational scoring systems and MAF to evaluate the pathogenicity of the BRCA variants found in their cohort. The approaches are all common and novelty of the study is limited.

Thank you for this comment. However, we believe that the use of the prospective cohort’s self-reporting data for validation of the pathogenicity of the variants in BRCA1/2 is quite a novel idea. 

2. It is not clear why the difference was observed only in the cancer-bearing sisters of the TMM CommCohort. It seems that the paper-based questionnaires is not so robust to differentiate the pathogenic BRCA variant carriers or to evaluate BRCA annotation systems.

Thank you for this comment, with which we basically agree. We did not intend to identify a new, highly penetrant pathogenic variant through this study. Instead, we attempted to develop a method to identify “moderately pathogenic” or “low penetrant” deleterious variants. To validate such variants, large-scale data collection methods are critical. Therefore, a paper-based questionnaire is appropriate to obtain the necessary information.

3. Functional analysis is recommended to confirm their annotation is accurate for the variants discordance was observed between ClinVar and their annotation system.

Thank you for the constructive comment. We performed some functional analyses to check this discordance. However, HBOC is not an acute disorder. Simple functional analysis may not be applicable for evaluating the small differences in the functions of the variants. Therefore, epidemiological methods would be more appropriate.

Minor comments are the following:

1. The meanings of sentence p20, l297-299 is not clear

Thank you for this constructive comment. Accordingly, we amended the text as follows:

“Four variants were annotated as “pathogenic” by Momozawa et al. [14], but not annotated as P or LP by InterVar (Supplementary Table 4). Among them, three variants showed high CADD_phred (24–35) and Eigen_raw scores (0.571–0.871) (Supplementary Table 4)”.

2. p21, l309 cBioPortal.

Thank you for this comment. We fixed the capitalization of this term throughout the main text.

3. The total number of all candidate in Figure 3 should be 27 considering male and female number.

Thank you for carefully checking the data. We fixed the figure accordingly.

4. Poor figure resolution.

Thank you for this comment. We generated high-resolution figures and substituted them for the old ones. We hope that the new figures have sufficient resolution.

6. PLOS authors have the option to publish the peer review history of their article (what does this mean?). If published, this will include your full peer review and any attached files.

Do you want your identity to be public for this peer review? For information about this choice, including consent withdrawal, please see our Privacy Policy.

Reviewer #1: No

Reviewer #2: No

---

## [Decision Letter · Decision Letter 1]

23 Nov 2020

PONE-D-20-21851R1

Novel candidates of pathogenic variants of the BRCA1 and BRCA2 genes from a dataset of 3,552 Japanese whole genomes (3.5KJPNv2)

PLOS ONE

Dear Dr. Yasuda,

Thank you for submitting your manuscript to PLOS ONE. After careful consideration, we feel that it has merit but does not fully meet PLOS ONE’s publication criteria as it currently stands. Therefore, we invite you to submit a revised version of the manuscript that addresses the points raised during the review process.

We look forward to receiving your revised manuscript.

Kind regards,

Yonglan Zheng

Academic Editor

PLOS ONE

Reviewers' comments:

Reviewer's Responses to Questions

**Comments to the Author**

1. If the authors have adequately addressed your comments raised in a previous round of review and you feel that this manuscript is now acceptable for publication, you may indicate that here to bypass the “Comments to the Author” section, enter your conflict of interest statement in the “Confidential to Editor” section, and submit your "Accept" recommendation.

Reviewer #1: (No Response)

Reviewer #2: All comments have been addressed

2. Is the manuscript technically sound, and do the data support the conclusions?

Reviewer #1: Partly

Reviewer #2: Yes

3. Has the statistical analysis been performed appropriately and rigorously? 

Reviewer #1: I Don't Know

Reviewer #2: Yes

4. Have the authors made all data underlying the findings in their manuscript fully available?

Reviewer #1: Yes

Reviewer #2: Yes

5. Is the manuscript presented in an intelligible fashion and written in standard English?

Reviewer #1: Yes

Reviewer #2: Yes

6. Review Comments to the Author

Reviewer #1: The authors’ efforts are appreciated with regard to their revisions on the manuscript. This has resulted in text that is significantly improved, but continues to have some ongoing issues towards publication.

MAJOR:

(1) With regard to InterVar use, to reiterate/clarify the point from the initial review, the authors did not explain their handling of noncoding variants (which is, as they clarified, is the point of WGS data over exonic data). Intervar describes handling of the mutation types in question as follows:

*Splicing -> Intervar does address some splicing variants, but they describe in the Li paper that their data cleaning procedure includes removal of variants with conflicting interpretation as part of data cleaning prior to checking the dbscsnv11 database. (This is confusing as the authors then show variants that have conflicting interpretations in ClinVar? I wasn't clear how these would have been identified if Intervar was used as described in the Li paper?)

*Intronic / noncoding variants -> This point was not addressed by the authors although the WGS would allow them to do so (and that is noted as one of their reasons for novelty). ANNOVAR identifies intergenic and noncoding variants, but InterVar’s web application is specifically designed for annotation use in exons and their paper notes that:

“InterVar is designed to interpret genetic variants that are likely to cause Mendelian diseases or are highly penetrant for Mendelian diseases (OR > 5) and cannot handle alleles that increase susceptibility to common and complex traits. Therefore, we caution that the current interpretation is appropriate only for Mendelian diseases or Mendelian forms of complex diseases.”

If the authors intend to only address splicing using WGS data over the existing exonic data, then region of application is what needs to be clarified consistently through the paper and a point should be made in the discussion regarding limitations of InterVar in this context.

If they intend to state that they analyzed intronic / noncoding variants related to BRCA1/BRCA2, then it is not clear that InterVar as described with use of default settings is a good tool for intronic / noncoding variants. I would refer the authors to the literature for references such as this: https://www.ncbi.nlm.nih.gov/pmc/articles/PMC6266896/ It is noted that the variant annotation, albeit different, cited by the authors in their prior analysis on 2KJPN demonstrates ability to pursue this work – this point is just not clear.

(2) There continue to be problems with conflation of moderate penetrance and likely pathogenic variants that are concerning. The term “moderately pathogenic variants” remains unclear. Please revise “moderately pathogenic” to “likely pathogenic” (related to variants) or “moderate penetrance” (related to genes) depending on what was intended as these are the standard accepted terms in the field.

For instance, these descriptions are completely inaccurate regarding management of patients with BRCA1/BRCA2 mutations because of this conflation problem.

“The carriers of moderately pathogenic HBOC variants will not have

undergone the drastic prophylactic modalities but frequent examination will be recommendable for earlier detection of the cancers.”

“The presence of such variants may not be critical to prompt radical interventions such as

prophylactic surgery, but the carriers may be encouraged to continue undergoing close health checks to detect HBOC cancers as early as possible.”

Also:

“Many of the cancer-predisposing genes are known to be associated with juvenile cancer

syndromes such as Li Fraumeni syndrome. The variants responsible for juvenile cancer syndromes

are usually very pathogenic and show strong effects on gene functions.”

Li Fraumeni syndrome is a specific diagnosis associated with a specific gene, not multiple genes. These sentences in particular should just be removed completely.

(3) If part of the purpose of the paper is to evaluate the performance of InterVar in annotation, ClinVar and InterVar variants should be treated as separate categories. There are multiple parts of the results in which variants from these groups are just put together.

MINOR:

Introduction:

-On page 4, in the introduction, the term “approaches” is unclear since what the authors are referring to is that there are several efforts (whereas approaches implies different strategies of ascertaining pathogenicity of identified mutations).

-Would modify result/discussion subtitles from “Pathogenic variants in the two…” -> “Estimation of pathogenic variants…” and “Estimate of computational scoring tools for pathogenicity of the 3.5KJPNv2 BRCA variants” -> “Estimate of computational scoring tools’ performance in predicting pathogenicity of novel 3.5KJPNv2 BRCA variants”

-Please revise references to genomAD for spelling/capitalization accuracy in the Introduction.

-Methods:

-The context for the statement in the methods regarding candidate variants in the Korean population is not clear.

-Continuing to have a lot of difficulty understanding how the authors were able to link information from the familial TMM database / questionnaire with the sequencing information if these are located in two separate datasets w/ two separate accesses.

-Results/Discussion

The authors’ note about cancer-bearing offspring mentions p=0.041 is not statistically significant but does not specify what threshold would be significant – presumably with a Chi-squared test this would be 0.05, so this is statistically significant?). However, if there was a correction done for multiple testing (which would be appropriate if testing was done across numerous variants), then this is not clear from the paper as written.

Reviewer #2: (No Response)

7. PLOS authors have the option to publish the peer review history of their article (what does this mean?). If published, this will include your full peer review and any attached files.

Reviewer #1: No

Reviewer #2: No

---

## [Author Response · Author response to Decision Letter 1]

2 Dec 2020

PONE-D-20-21851R1

Novel candidates of pathogenic variants of the BRCA1 and BRCA2 genes from a dataset of 3,552 Japanese whole genomes (3.5KJPNv2)

6. Review Comments to the Author

Reviewer #1: The authors’ efforts are appreciated with regard to their revisions on the manuscript. This has resulted in text that is significantly improved, but continues to have some ongoing issues towards publication.

MAJOR: (1) With regard to InterVar use, to reiterate/clarify the point from the initial review, the authors did not explain their handling of noncoding variants (which is, as they clarified, is the point of WGS data over exonic data). Intervar describes handling of the mutation types in question as follows:

*Splicing -> Intervar does address some splicing variants, but they describe in the Li paper that their data cleaning procedure includes removal of variants with conflicting interpretation as part of data cleaning prior to checking the dbscsnv11 database. (This is confusing as the authors then show variants that have conflicting interpretations in Clinvar? I wasn't clear how these would have been identified if Intervar was used as described in the Li paper?)

Thank you very much for detailed comment. To answer to the point, we checked the Intervar annotation data again and we did not find any variants that may affect splicing but interpreted as “Benign” or “likely benign” in the BRCA1/2 gene variants in 3.5KJPN. There is one variant, the BRCA2 I2675V, that is known as a splicing error-causing variant in 3.5KJPN and the variant is annotated as P or LP by both Clinvar and Intervar. To clarify the issue, we added following sentences in the “Results and discussions” section (p22).

 “So far there is no other splicing error-causing variants of the BRCA1/2 genes in the 3.5KJPN. As shown in Table 1, there are two splicing-affected variants (BRCA2:g.chr13: 32890558AGdel and BRCA2:g.chr13: 32937315G>A) in the GnomAD-EAS, indicating that we did not miss a large numbers of splicing error causing variants in the BRCA1/2 genes in the 3.5KJPN.” 

*Intronic / noncoding variants -> This point was not addressed by the authors although the WGS would allow them to do so (and that is noted as one of their reasons for novelty). ANNOVAR identifies intergenic and noncoding variants, but InterVar’s web application is specifically designed for annotation use in exons and their paper notes that:

“InterVar is designed to interpret genetic variants that are likely to cause Mendelian diseases or are highly penetrant for Mendelian diseases (OR > 5) and cannot handle alleles that increase susceptibility to common and complex traits. Therefore, we caution that the current interpretation is appropriate only for Mendelian diseases or Mendelian forms of complex diseases.”

If the authors intend to only address splicing using WGS data over the existing exonic data, then region of application is what needs to be clarified consistently through the paper and a point should be made in the discussion regarding limitations of InterVar in this context.

If they intend to state that they analyzed intronic / noncoding variants related to BRCA1/BRCA2, then it is not clear that InterVar as described with use of default settings is a good tool for intronic / noncoding variants. I would refer the authors to the literature for references such as this: https://www.ncbi.nlm.nih.gov/pmc/articles/PMC6266896/ It is noted that the variant annotation, albeit different, cited by the authors in their prior analysis on 2KJPN demonstrates ability to pursue this work – this point is just not clear.

Thank you very much for the comment and we apologize our unclear explanations. We have used command-line based InterVar software with its default option. It can annotate the noncoding variants of a gene and intergenic regions, too. As we mentioned in the methods section of our manuscript, the command-line Intervar is depends on the ANNOVAR function so that it can cover non-coding variants. In the BRCA1/2 genes in the 3.5KJPN, there are 2891 non-coding (intronic, 5’-UTR, and 3’-UTR) variants and all of them are annotated by the software. There is no P or LP variants in either Clinvar or Intervar. To clarify the issue, we added the following sentences in the “Results and discussions” section (p17).

Hence, InterVar may underestimate the clinical impact of potentially pathogenic variants about which previous information is not available. The tendency might be worse in the noncoding regions in the coding genes like the BRCA1/2 genes because of the lack of functional studies for such regions. Nowadays, the whole genome sequencing data is accumulating and comparisons between the phenotypes and variants in the noncoding regions found by the WGS will provide critical data for the interpretation of the noncoding variants.” 

(2) There continue to be problems with conflation of moderate penetrance and likely pathogenic variants that are concerning. The term “moderately pathogenic variants” remains unclear. Please revise “moderately pathogenic” to “likely pathogenic” (related to variants) or “moderate penetrance” (related to genes) depending on what was intended as these are the standard accepted terms in the field.

For instance, these descriptions are completely inaccurate regarding management of patients with BRCA1/BRCA2 mutations because of this conflation problem.

“The carriers of moderately pathogenic HBOC variants will not have

undergone the drastic prophylactic modalities but frequent examination will be recommendable for earlier detection of the cancers.”

“The presence of such variants may not be critical to prompt radical interventions such as

prophylactic surgery, but the carriers may be encouraged to continue undergoing close health checks to detect HBOC cancers as early as possible.”

Thank you very much for the comments. We understand the reviewer’s criticism as a word “pathogenic” should not be used for a variant with low susceptibility that may not be needed any medical drastic actions. We would like to use “likely pathogenic” for variants that can cause diseases with enough high provability for medical action and “moderate” variants that shows low but significantly susceptible for disease onset. So, we amended the issue throughout the manuscript not to use “pathogenic” for explanation of such weakly disease-causing variants. In terms of the two exemplified sentences, we amended as follows:

“The carriers of moderately deleterious HBOC variants would not undergo drastic prophylactic modalities, but frequent examination would be recommendable for earlier detection of the cancers.”

“The carriers in moderately penetrant HBOC families may not be critical to prompt radical interventions such as prophylactic surgery, but the carriers may be encouraged to continue undergoing close health checks to detect HBOC cancers as early as possible.”

Also:

“Many of the cancer-predisposing genes are known to be associated with juvenile cancer

syndromes such as Li Fraumeni syndrome. The variants responsible for juvenile cancer syndromes

are usually very pathogenic and show strong effects on gene functions.”

Li Fraumeni syndrome is a specific diagnosis associated with a specific gene, not multiple genes. These sentences in particular should just be removed completely.

Thank you very much for critical comment and we accept the request and deleted the sentences.

(3) If part of the purpose of the paper is to evaluate the performance of InterVar in annotation, Clinvar and InterVar variants should be treated as separate categories. There are multiple parts of the results in which variants from these groups are just put together.

Thank you very much for the careful comments and we apologize the confusing way of our explanation. We did not include the evaluation of performance of InterVar in annotation. We just use it to supplement the lack of annotation by Clinvar and obtain the info of the other parameters such as CADD, DAN, and Eigen. Most of the conflicts between two databases are caused by the lack of information in Clinvar. So, we use them basically the same manner.

MINOR:

Introduction:

-On page 4, in the introduction, the term “approaches” is unclear since what the authors are referring to is that there are several efforts (whereas approaches implies different strategies of ascertaining pathogenicity of identified mutations).

Thank you very much for careful comments and we amended the sentences to clarify our intention as follows:

“Several levels of studies (single organization, single nation, and international level) have been done previously. As a single organization study, Sugano et al. reported the BRCA1 and BRCA2 germline variants in 135 HBOC patients and identified 28 pathogenic ones [9]. As the nationwide study, Arai et al. examined 830 Japanese HBOC pedigrees collected by the Japanese HBOC consortium and identified 49 different pathogenic variants among them [10]. Similarly, a nationwide multicenter study revealed that germline BRCA 1/2 mutations were present in 14.7% of 634 Japanese women with ovarian cancer [5]. Lee et al. also examined the variants in the BRCA1 and BRCA2 genes in breast and ovarian cancer patients’ germline genomic DNA and calculated posterior probabilities for the disease-causing mutations; they identified five previously unreported variants as candidate pathogenic ones [11]. Finally, as an international study, BRCA Exchange…”

-Would modify result/discussion subtitles from “Pathogenic variants in the two…” -> “Estimation of pathogenic variants…” and “Estimate of computational scoring tools for pathogenicity of the 3.5KJPNv2 BRCA variants” -> “Estimate of computational scoring tools’ performance in predicting pathogenicity of novel 3.5KJPNv2 BRCA variants”

Thank you very much for critical comment and we accept the request and amended as suggested. 

-Please revise references to genomAD for spelling/capitalization accuracy in the Introduction.

 Thank you so much for the careful comment and we amended the issue throughout the manuscript as “gnomAD”.

-Methods:

-The context for the statement in the methods regarding candidate variants in the Korean population is not clear.

Thank you very much for the comment. We tried to fix the issue as following:

“The positions of the candidate pathological variants found in the Korean population [11] were described as the cDNA positions. Ti apply the data to the InterVar software, …“

-Continuing to have a lot of difficulty understanding how the authors were able to link information from the familial TMM database / questionnaire with the sequencing information if these are located in two separate datasets w/ two separate accesses.

Thank you for the comment and we apologize the unclear explanation. So, we added the following explanation in the Methods section. 

“The TMM database is a relational database and it consists of several separate datasets. The key is the participants’ IDs to link the information stored in the different tables”

-Results/Discussion

The authors’ note about cancer-bearing offspring mentions p=0.041 is not statistically significant but does not specify what threshold would be significant – presumably with a Chi-squared test this would be 0.05, so this is statistically significant?). However, if there was a correction done for multiple testing (which would be appropriate if testing was done across numerous variants), then this is not clear from the paper as written.

Thank you very much for the constructive comments. We felt that the 0.041 was not enough low to be “significant”. As the reviewer suggested, it should be interpreted as “marginally significant” if the p < 0.05 is the threshold. So we amended the main text as follows. To fulfil the requirement of the PLoS One policy for publication, we added the data for offspring cancer burden in the TMM database as Supplementary Table 5 and fixed Figure 2 accordingly. So, the figure legend of Figure 2 also amended.

“A prominent difference between those definitely carrying potentially pathogenic BRCA variants and the rest of the cohort was in the rate of cancer-bearing sisters: the InterVar P or LP carriers were shown to have a much higher rate of cancer-bearing sisters than the rest of the cohort (Fig. 2 and Supplementary Table 5; p = 3.08 × 10−5, chi-squared test with Yates’ correction). In addition, the rate of cancer-bearing offspring was higher in the InterVar P or LP carriers than in the others with marginally significant (p = 0.041).”

Figure 2 legend: 

“…Asterisks indicate statistically significant differences (single: p < 0.05, double: p < 10-4) upon comparison with the total analyzed TMM CommCohort cases (Fig. 2).”

Reviewer #2: (No Response)

7. PLOS authors have the option to publish the peer review history of their article (what does this mean?). If published, this will include your full peer review and any attached files.

Do you want your identity to be public for this peer review? For information about this choice, including consent withdrawal, please see our Privacy Policy.

Reviewer #1: No

Reviewer #2: No

---

## [Editor Report · Decision Letter 2]

7 Dec 2020

Novel candidates of pathogenic variants of the BRCA1 and BRCA2 genes from a dataset of 3,552 Japanese whole genomes (3.5KJPNv2)

PONE-D-20-21851R2

Dear Dr. Yasuda,

We’re pleased to inform you that your manuscript has been judged scientifically suitable for publication and will be formally accepted for publication once it meets all outstanding technical requirements.

Kind regards,

Yonglan Zheng

Academic Editor

PLOS ONE

---

## [Editor Report · Acceptance letter]

22 Dec 2020

PONE-D-20-21851R2 

Novel candidates of pathogenic variants of the *BRCA1* and *BRCA2* genes from a dataset of 3,552 Japanese whole genomes (3.5KJPNv2) 

Dear Dr. Yasuda:

I'm pleased to inform you that your manuscript has been deemed suitable for publication in PLOS ONE. Congratulations! Your manuscript is now with our production department. 

Kind regards, 

on behalf of

Dr. Yonglan Zheng 

Academic Editor

PLOS ONE